# AR activates YAP/TAZ differentially in prostate cancer

Omar Salem[1,2], Siyang Jia[1,2], Bin-Zhi Qian[3], Carsten Gram Hansen[1,2]

The Hippo signalling pathway is a master regulator of cell growth, proliferation, and cancer. The transcriptional coregulators of the Hippo pathway, YAP and TAZ, are central in various cancers. However, how YAP and TAZ get activated in most types of cancers is not well understood. Here, we show that androgens activate YAP/TAZ via the androgen receptor (AR) in prostate cancer (PCa), and that this activation is differential. AR regulates YAP translation while inducing transcription of the TAZ encoding gene, *WWTR1*. Furthermore, we show that AR-mediated YAP/TAZ activation is regulated by the RhoA GTPases transcriptional mediator, serum response factor (SRF). Importantly, in prostate cancer patients, *SRF* expression positively correlates with *TAZ* and the YAP/TAZ target genes *CYR61* and *CTGF*. We demonstrate that YAP/TAZ are not essential for sustaining AR activity, however, targeting YAP/TAZ or SRF sensitize PCa cells to AR inhibition in anchorage-independent growth conditions. Our findings dissect the cellular roles of YAP, TAZ, and SRF in prostate cancer cells. Our data emphasize the interplay between these transcriptional regulators and their roles in prostate tumorigenesis and highlight how these insights might be exploited therapeutically.

## Introduction

Prostate cancer (PCa) is a leading cause of male's mortality globally. Annually, more than one million new cases are reported (Wong et al, 2016; Bray et al, 2018). Aberrant androgen receptor (AR) signalling is a potent promoter of PCa development, progression, and metastasis (Scher & Sawyers, 2005; Mohler, 2008; Messner et al, 2020). AR is a transcription factor that belongs to the superfamily of steroid hormone receptors (Modi et al, 2016). Inactive AR resides in the cytoplasm, where AR complexes with heat shock proteins such as HSP90. Upon binding of dihydrotestosterone (DHT), AR dissociates from the heat shock proteins, translocates to the nucleus, and binds to androgen response elements (ARE) to induce the expression of multiple target genes (Modi et al, 2016).

AR amplifications and gain of function mutations (Brooke & Bevan, 2009), and altered expression of AR co-regulators, render AR hyperactive (Culig & Santer, 2012). These various forms of dysregulated AR cause AR hyperactivity, which is apparent across several stages of PCa (Messner et al, 2020). Targeting AR via androgen deprivation therapy continues to be the standard line of treatment against hormone responsive PCa (Chen et al, 2008; Coutinho et al, 2016). However, in general this is a temporary solution, as unfortunately, many patients develop castration-resistant prostate cancer (CRPC). Molecular evidence suggests that AR activity persists in CRPC even in the presence of enzalutamide, a second-generation AR inhibitor (Antonarakis et al, 2014). In this sense, there are limited long-term clinical benefits in targeting AR. There is therefore an imminent and substantial clinical need to understand the disease development in detail, including through exploring novel signalling pathways. These new insights might provide novel therapeutic targets and therefore new improved clinical opportunities.

The Hippo pathway is a principal regulator of cell growth, proliferation, and development (Riley et al, 2022). The Hippo pathway is tightly regulated by multiple upstream signals including cell density, polarity, mechanotransduction, nutrients, and via a range of G-protein-coupled receptors (Yu et al, 2015; Santinon et al, 2016; Totaro et al, 2018; Rausch et al, 2019). The canonical Hippo pathway comprises of an upstream serine–threonine kinase cascade that phosphorylates and thereby activates large tumour suppressor kinase (LATS1/2), which in turn phosphorylates and inhibits the downstream co-transcriptional regulators: Yes-associated protein (YAP) (Zhao et al, 2007) and its paralog: the transcriptional coactivator with PDZ-binding motif (TAZ) (Lei et al, 2008). YAP and TAZ do not bind to DNA directly and consequently make use of transcription factors to regulate gene expression, most notably via the TEADs. Notably, YAP and TAZ when binding to TEAD and potentially other transcription factors both activate and repress specific gene-sets, and as such can both function as either gene-specific co-activators or co-repressors (Zhao et al, 2008; Hansen et al, 2015a; Kim et al, 2015a; Walko et al, 2017; Kowalczyk et al, 2022). When the upstream kinase module of the Hippo pathway is engaged, this leads to LATS1/2-mediated YAP and TAZ phosphorylation on multiple serine sites. These phosphorylations render YAP and TAZ prone to cytoplasmic sequestration or degradation (Hansen et al, 2015a; Meng et al, 2016). Notably, kinase cascade-

[1]The University of Edinburgh, Centre for Inflammation Research, Edinburgh BioQuarter, Edinburgh, UK    [2]Institute for Regeneration and Repair, The University of Edinburgh, Edinburgh BioQuarter, Edinburgh, UK    [3]Medical Research Council Centre for Reproductive Health, College of Medicine and Veterinary Medicine, Queen's Medical Research Institute, The University of Edinburgh, Edinburgh, UK

Correspondence: carsten.g.hansen@ed.ac.uk

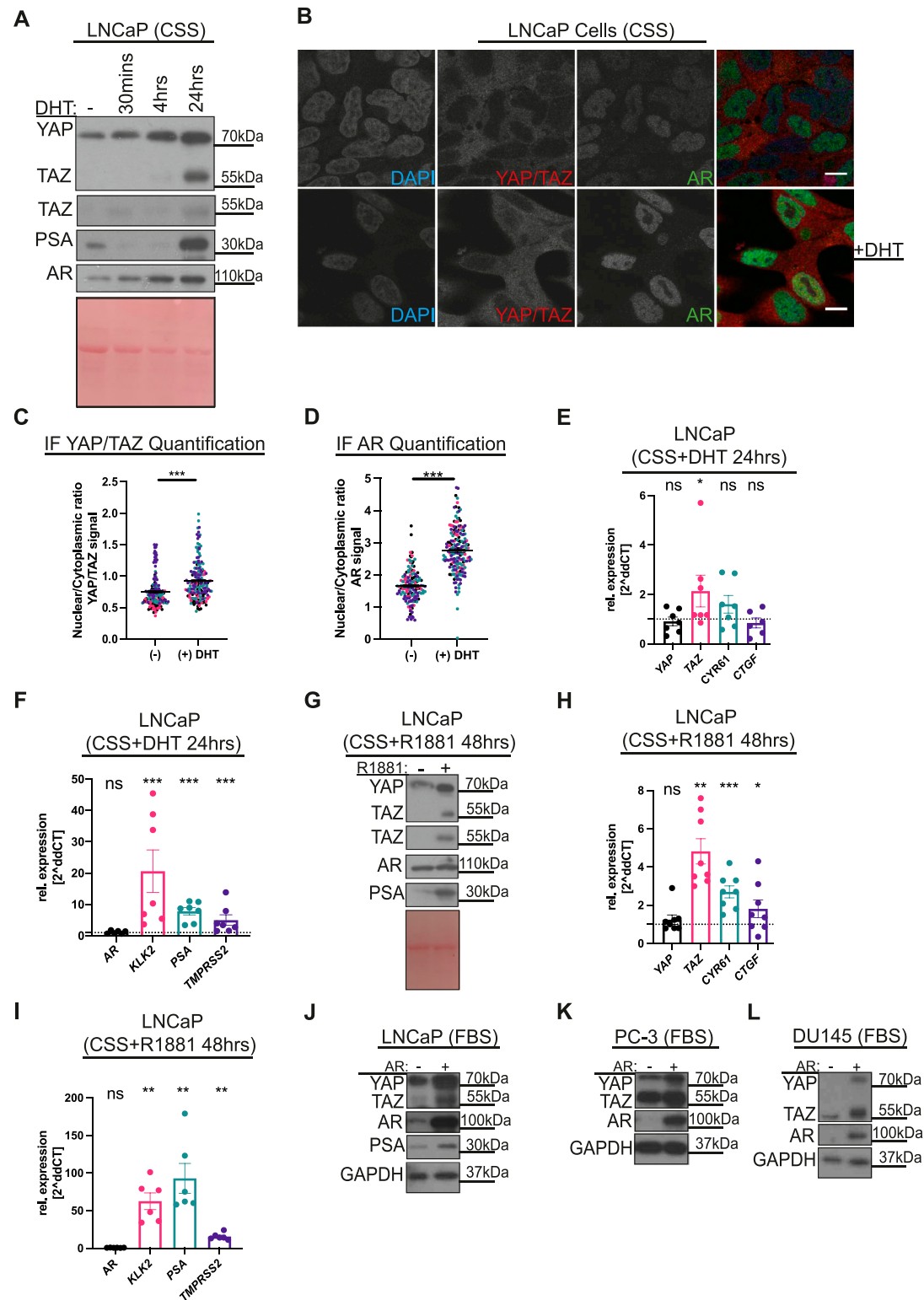

**Figure 1.   Androgens activate YAP/TAZ.**
**(A)** Western blots from LNCaP lysates. Cells were cultured in charcoal-stripped serum (CSS) conditions for 48 h and stimulated with vehicle (DMSO) or androgen (10 nM, dihydrotestosterone, DHT) for 30 min, 4 h, and 24 h before lysis. Ponceau S total protein stain serves as a loading control. **(B)** Confocal images of LNCaP cells stimulated with vehicle (DMSO), top row or androgen (10 nM, dihydrotestosterone, DHT) for 24 h in CSS conditions, bottom row. DAPI (blue), YAP/TAZ (red), and AR (green). Brightness and contrast of the merged images were enhanced to allow visualisation of the different proteins. Scalebar = 30 μm. **(C, D)** Dot plot of quantified YAP/TAZ and AR nuclear/cytoplasmic levels respectively from images, as shown in (B). Each dot represents one cell. Colour coding of dots highlight data from the same day. Mean ± SEM,

independent regulation of YAP/TAZ has also been reported (Chan et al, 2011; Dupont et al, 2011; Kofler et al, 2018).

YAP/TAZ is central in most solid tumours (Steinhardt et al, 2008; Moroishi et al, 2015a; Zanconato et al, 2016). In recent years, the implications of elevated activity of YAP/TAZ in PCa have gained attention (Salem & Hansen, 2019). Reports suggest that AR and YAP co-localise and interact in the nucleus to promote cancer progression and that this interaction is critical for AR function (Kuser-Abali et al, 2015). However, the exact mechanism of YAP activation in PCa remains unknown. Importantly, the role of how androgens and AR regulate TAZ (encoded by *WWTR1*) in PCa is largely unexplored. Likewise, overlapping and divergent YAP/TAZ functions and regulations are, in general, not well understood.

Here we, through multiple lines of evidence, show that androgens via AR activate YAP/TAZ differentially, where AR primarily regulates YAP translation while inducing transcription of the TAZ-encoding gene, *WWTR1*. RhoA and serum response factor (SRF) regulate this AR-driven YAP/TAZ activation in a feed forward manner. We highlight that YAP/TAZ are not essential for AR nuclear translocation and transcriptional activation, however, targeting YAP/TAZ or SRF provides a new promising therapeutic avenue to develop to manage prostate tumorigenesis.

# Results

### Androgens activate YAP/TAZ

Conventional tissue culture media contain FBS, and include growth factors, hormones, and metabolites, some of which engage AR. Consequently, there is a steady state AR output in PCa cells cultured in an FBS-containing medium (Cao et al, 2009). Therefore, to study the full extent of the AR downstream activity, cells were temporarily cultured in media containing charcoal stripped serum (CSS), allowing the disengagement of any ligands that are bound to AR. Thereafter, cells were stimulated with DHT for varying times (30 min, 4 and 24 h). DHT exposure robustly increases YAP/TAZ protein levels by 24 h (Fig 1A). YAP/TAZ levels positively correlate with AR activity, as indicated by the parallel increase in AR and the AR target prostate-specific antigen (PSA) levels (Fig 1A). In addition, immunofluorescence (IF) microscopy using separate antibodies recognising both YAP/TAZ (Hansen et al, 2015b; Rausch et al, 2019) and AR reveal that DHT-stimulated cells favour YAP/TAZ nuclear localisation relative to untreated cells (Fig 1B–D). Data from these complimentary approaches are consistent with that DHT activates YAP/TAZ.

We next sought to ask whether the observed apparent androgen-driven activation of YAP/TAZ (Fig 1A and B) is transcriptionally mediated and whether YAP/TAZ activation induces downstream gene expression. Surprisingly, qRT-PCR analysis carried out in LNCaP cells stimulated with DHT for 24 h revealed that *TAZ/WWTR1* is transcriptionally regulated by DHT, with no recorded induction of *YAP* or the common YAP/TAZ target genes *CYR61* nor *CTGF* (Fig 1E) at this timepoint. To confirm downstream AR activity, the expression of the canonical AR targets, *KLK2* (Sun et al, 1997), *PSA* (Schuur et al, 1996), and *TMPRSS2* (Wang et al, 2007) was analysed after DHT stimulation in parallel, all of which are, as expected, induced (Fig 1F).

Because we did not observe YAP/TAZ-induced transcriptional activity after 24 h of androgen stimulation, we hypothesised that YAP/TAZ proteins are being synthesised, and/or stabilised, during the initial period of androgen stimulation (Fig 1A) and that a longer term AR induction might be required to detect YAP/TAZ downstream transcriptional activity. For the purpose of prolonged AR stimulation, we used the synthetic androgen methyltrienolone (R1881), which has a 1.5–twofold higher affinity than DHT, and is not metabolised by LNCaP cells (Brown et al, 1981). LNCaP cells were stimulated with R1881 for 48 h, whereafter, cell lysates were analysed. These data show that elevated YAP/TAZ protein and *TAZ/WWTR1* mRNA expression were maintained upon androgen stimulation at this later time point (Fig 1G and H). We next analysed the expression of YAP/TAZ target genes. Consistent with our hypothesis that prolonged androgen stimulation would induce YAP/TAZ target genes, *CYR61* and *CTGF* are induced after 48 h R1881 stimulation (Fig 1H). However, *YAP* mRNA levels remain unchanged (Fig 1H). Notably, the increased expression of *KLK2*, *PSA*, and *TMPRSS2* is maintained after 48 h AR induction (Fig 1I). Complimentary to the ligand stimulation, we next asked whether exogenous AR expression induce YAP/TAZ in LNCaP, and in the AR-negative cell lines, PC-3 and DU145. Plasmid-based exogenous AR expression was confirmed using Western blotting analysis (Fig 1J–L). Importantly, by probing the same cell lysates against YAP/TAZ, we noted that this elevated AR expression positively correlates with YAP/TAZ across all three cell lines (Fig 1J–L).

### Differential YAP/TAZ activation by AR

To confirm that DHT and R1881 effects on YAP/TAZ are mediated via AR, we targeted *AR* with shRNA. AR knockdown was confirmed by probing for AR and PSA by Western blots and measuring *AR* gene expression by qRT-PCR (Fig 2A and B). AR loss results in decreased YAP/TAZ protein levels (Fig 2A). Surprisingly, upon AR knockdown, *YAP* mRNA is increased, whereas *TAZ(WWTR1)* mRNA levels are

C and D were analysed with *t* test and Mann–Whitney *U* test, respectively. **(E)** qRT-PCR analysis of *YAP*/TAZ(*WWTR1*) and established YAP/TAZ targets. mRNA from LNCaP stimulated with androgen (10 nM, dihydrotestosterone, DHT) for 24 h in CSS conditions compared with vehicle (DMSO) in CSS conditions. Each dot represents data from a biological replicate. Mean ± SEM, Mann–Whitney *U* test. **(F)** qRT-PCR as shown in (E) analysed for the expression of AR and established AR target genes. Each dot represents data from a biological replicate. Mean ± SEM, Mann–Whitney *U* test. **(G)** Western blots of lysates from LNCaP cells. Cells were cultured in CSS for 48 h and treated with vehicle (DMSO) or androgen (1 μM, R1881) for 48 h. **(H)** qRT-PCR analysis of *YAP*/TAZ(*WWTR1*) and established YAP/TAZ targets. mRNA from LNCaP treated with androgen (1 μM, R1881) for 48 h in CSS conditions compared with vehicle (DMSO). Each dot represents data from a biological replicate. Mean ± SEM, Mann–Whitney *U* test. **(I)** qRT-PCR as shown in (H) analysed for AR expression and established AR target genes. Each dot represents data from a biological replicate. Mean ± SEM, Mann–Whitney *U* test. **(J)** Western blot lysates from LNCaP cells −/+ exogenously expressing AR. **(K)** Western blot lysates from PC-3 cells −/+ exogenously expressing AR. **(L)** Western blot lysates from DU145 cells −/+ exogenously expressing AR.

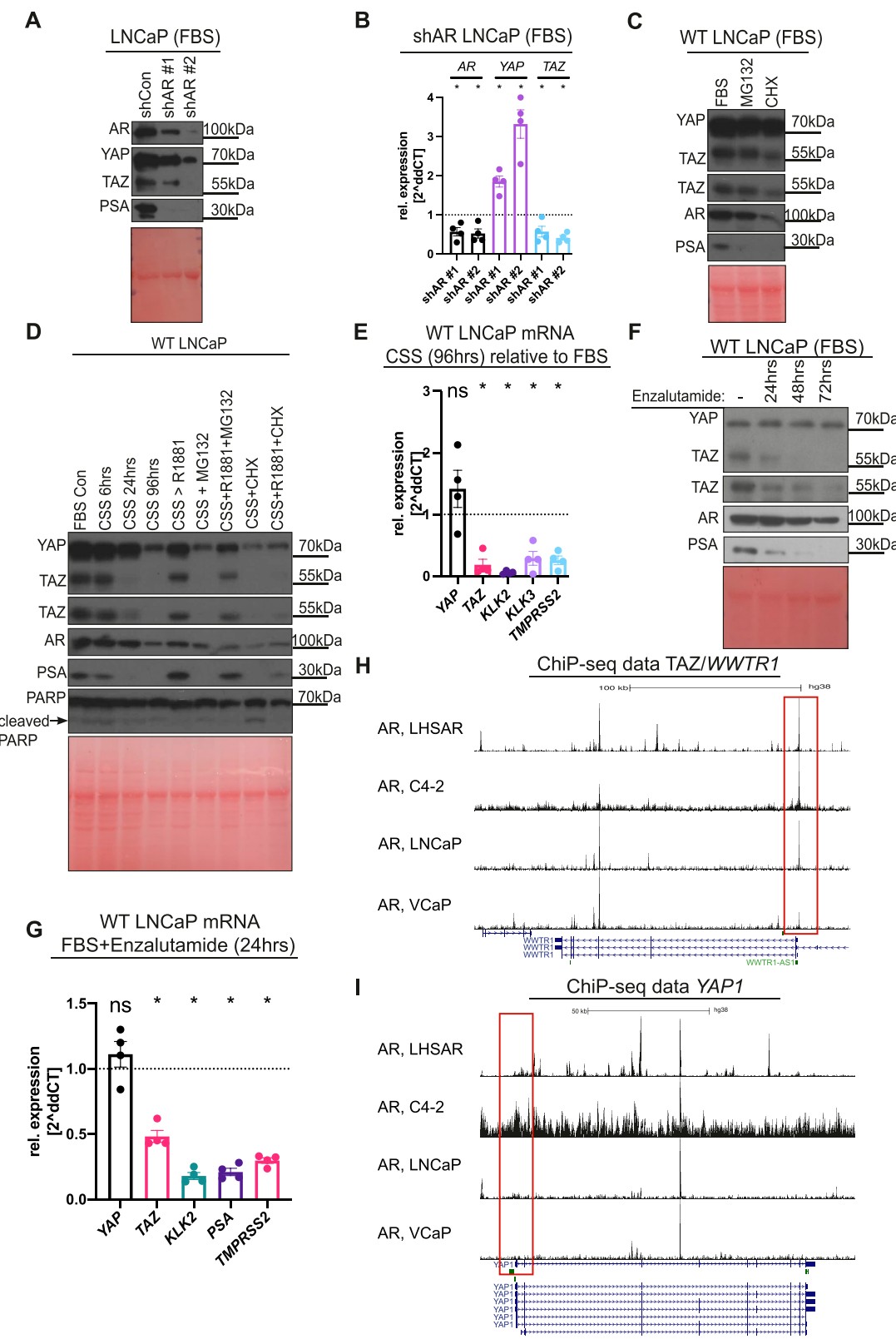

**Figure 2. Differential YAP/TAZ activation by AR.**
**(A)** Western blot analysis of lysates from LNCaP cells cultured in FBS expressing whether shCon or two separate short hairpin (sh) sequences targeting AR mRNA (shAR #1 and #2). **(B)** qRT-PCR analysis of *AR, YAP, TAZ(WWTR1)* mRNA from shAR LNCaP cells compared with shCon LNCaP cells. Each dot represents data from a biological replicate. Mean ± SEM, Mann–Whitney *U* test. **(C)** Western blot analysis of lysates from LNCaP cells cultured in FBS conditions. Cells were treated with proteosome inhibitor

decreased (Fig 2A and B), further highlighting the differential transcriptional regulation of YAP and TAZ (Fig 1E and H). We hypothesise that *YAP* mRNA up-regulation upon AR loss (Fig 2B) is likely a compensatory mechanism by which cells attempt to replenish decreased levels of YAP (and TAZ). We provide evidence indicating that AR regulates *TAZ* transcriptionally while modulating YAP protein levels. We next sought to analyse if AR induces YAP protein synthesis or inhibits YAP proteasomal degradation. We therefore used the translation inhibitor cycloheximide (CHX) and the proteosome inhibitor (MG132) to investigate these processes further.

First, we tested the effects of MG132 and CHX on LNCaP cells at steady state, both of which appeared to have no effect on YAP/TAZ protein levels (Fig 2C). Next, we measured YAP/TAZ protein levels in response to CSS-mediated AR inhibition. To understand protein level dynamics, FBS media was substituted with CSS media for varying intervals (6, 24, and 96 h). CSS treatment inhibit as expected AR and lowered PSA protein levels, confirming inhibition of AR activity (Fig 2D). This effect is correlated with decrease in YAP/TAZ protein levels (Fig 2D). AR, PSA, and YAP/TAZ loss is rescued by R1881 stimulation for 48 h (Fig 2D). This further confirms that androgens are sufficient to increase YAP/TAZ levels even in the absence of various FBS components not present in CCS (Yu et al, 2012). To examine whether AR regulates YAP/TAZ via proteasomal degradation, LNCaP cells in CSS conditions were either treated with vehicle for 48 h and MG132 for 6 h or R1881 for 48 h and MG132 for 6 h. Notably, proteasome inhibition by MG132 did not suppress YAP/TAZ, AR or PSA induction by R1881, excluding the role of proteasomes in this regulatory mechanism (Fig 2D). To test whether AR regulates the translation of YAP/TAZ, similar conditions were used in parallel, where LNCaP cells in CSS conditions were treated with vehicle for 48 h and CHX for 6 h or R1881 for 48 h and CHX for 6 h. Inhibiting translation by CHX in LNCaP cells for 6 h causes loss of R1881-induced YAP/TAZ activation. These data establish that AR regulates the translation of YAP/TAZ (Fig 2C and D).

To gain further insights into whether inhibition of YAP/TAZ due to CSS treatment is transcriptionally mediated, we performed qRT-PCR on LNCaP cells cultured in CSS conditions for 96 h relative to FBS conditions. Consistently, CSS inhibits *TAZ (WWTR1)*, and the established AR target genes *KLK2*, *PSA*, and *TMPRSS2*, without affecting *YAP* mRNA (Fig 2E).

As a complimentary method to inhibit AR, we took advantage of the clinically used second-generation AR inhibitor, enzalutamide (Hussain et al, 2018). Enzalutamide was added to LNCaP cells cultured in full FBS media for different time points (24, 48, and 72 h). We confirmed AR inhibition in response to enzalutamide by blotting against AR and PSA (Fig 2F). We also analysed AR target gene expression and observed as expected a decrease in *KLK2*, *PSA*, and

*TMPRSS2* expressions (Fig 2G). Enzalutamide results in decreased TAZ protein expression without decreasing YAP levels (Fig 2F). Consistent with AR knockdown and steroid deprivation by CSS, enzalutamide inhibits *TAZ(WWTR1)* transcriptionally without affecting *YAP* when analysed at 24 h of enzalutamide treatment (Fig 2G). Our results indicate that AR regulates YAP and TAZ differentially. Our data combined show that AR controls YAP translation without affecting *YAP* gene expression. Conversely, *TAZ(WWTR1)* is transcriptionally induced upon AR activation, which consequently causes an increase in TAZ protein. To further confirm our results, we analysed ChiP-Seq data for *WWTR1* and *YAP1* from the Cistrome database (Liu et al, 2011). These analyses indicate the presence of IP peaks in the DNA region presumably containing the *TAZ (WWTR1)* promoter in a range of PCa cell lines (LHSAR, C4-2, LNCaP, and VCaP) (Fig 2H), a binding that is not recorded when analysing the *YAP1* promoter region (Fig 2I).

## AR-mediated YAP/TAZ activation is regulated by the RhoA–SRF signalling axis

Previous studies revealed that RhoA responds to androgens and that androgens induce GTPase-dependent transcription (Schmidt et al, 2012). Because YAP and TAZ respond to RhoGTPase activation and change in the actin cytoskeleton tension (Yu et al, 2012; Zhao et al, 2012; Aragona et al, 2013; Moroishi et al, 2015b; Park et al, 2015; Mason et al, 2019), we examined whether RhoA (Sahai et al, 1998; Hodge & Ridley, 2016) is part of the signalling nexus between AR and YAP/TAZ. We used shRNA to target *RhoA*. RhoA loss leads to decreased YAP/TAZ protein (Fig 3A), *TAZ (WWTR1)* mRNA levels, and inhibition of YAP/TAZ-mediated gene transcription (Figs 3B and S1C). We next asked whether AR is able to override RhoA-mediated inhibition of YAP/TAZ. Exogenously expressing AR in RhoA KD LNCaP cells restore YAP/TAZ protein levels and *TAZ(WWTR1)*, *CYR61*, and *CTGF* mRNA, without effecting *YAP* mRNA levels (Fig 3A and C). In addition, we observe that exogenous expression of the dominant negative RhoA N19 (Ren, 1999) in LNCaP cells inhibits YAP nuclear translocation after DHT treatment (Figs 3D and E).

To further dissect RhoA responses, we aimed to identify a *WWTR1*/TAZ transcriptional regulator that is downstream of RhoA GTPases and is dependent of AR signalling. Previous studies highlight that SRF is a potential transcriptional regulator of *TAZ(WWTR1)* in breast cancer (Liu et al, 2016). Importantly, SRF interacts with YAP in mammary epithelial and breast cancer cells to drive cancer stemness (Kim et al, 2015b). Although breast and PCas are physiological and anatomically different, both require steroids for development, are glandular, and their tumorigenesis often leads to the development of hormone-regulated cancers (Risbridger et al,

---

(10 µM, MG132) or translation inhibitor cycloheximide (100 µg/ml, CHX) for 6 h. **(D)** Western blot analysis of LNCaP cells. Control cells were cultured in FBS for the full experimental period labelled "FBS Con" and compared with cells treated with CSS for a time course of 6 h, 24 h or 96 h. Cells were further treated with androgen (1 µm, R1881) or 10 µM, MG132, R1881+MG132, 100 µg/ml, CHX or R1881+ cycloheximide for the indicated durations. Note the loss of R1881-mediated induction of YAP/TAZ/PSA upon CHX treatment. **(E)** qRT-PCR analysis of *YAP/TAZ(WWTR1)* and established AR targets, *KLK2*, *PSA*, and *TMPRSS2*. mRNA from LNCaP cells in CSS conditions for 96 h compared with FBS conditions from the same time point. Mean ± SEM Mann–Whitney *U* test. **(F)** Western blot analysis of lysates from LNCaP cells treated with AR antagonist (10 µM, enzalutamide) for 24 h, 48 h or 72 h in FBS. Note the differential response of YAP/TAZ upon AR inhibition. **(G)** qRT-PCR analysis of *YAP/TAZ*, *AR*, and AR targets. mRNA from LNCaP treated with enzalutamide for 24 h compared with vehicle (DMSO) in FBS. Each dot represents data from a biological replicate. Mean ± SEM, Mann–Whitney *U* test. **(H, I)** AR binds to regions of *WWTR1* (encoding TAZ) but not the *YAP1* promoter. ChiP-seq data obtained from the cistrome database across four cell lines (LHSAR, C4-2, LNCaP, and VCAP) for *WWTR1* (top) and *YAP1* (bottom), respectively.

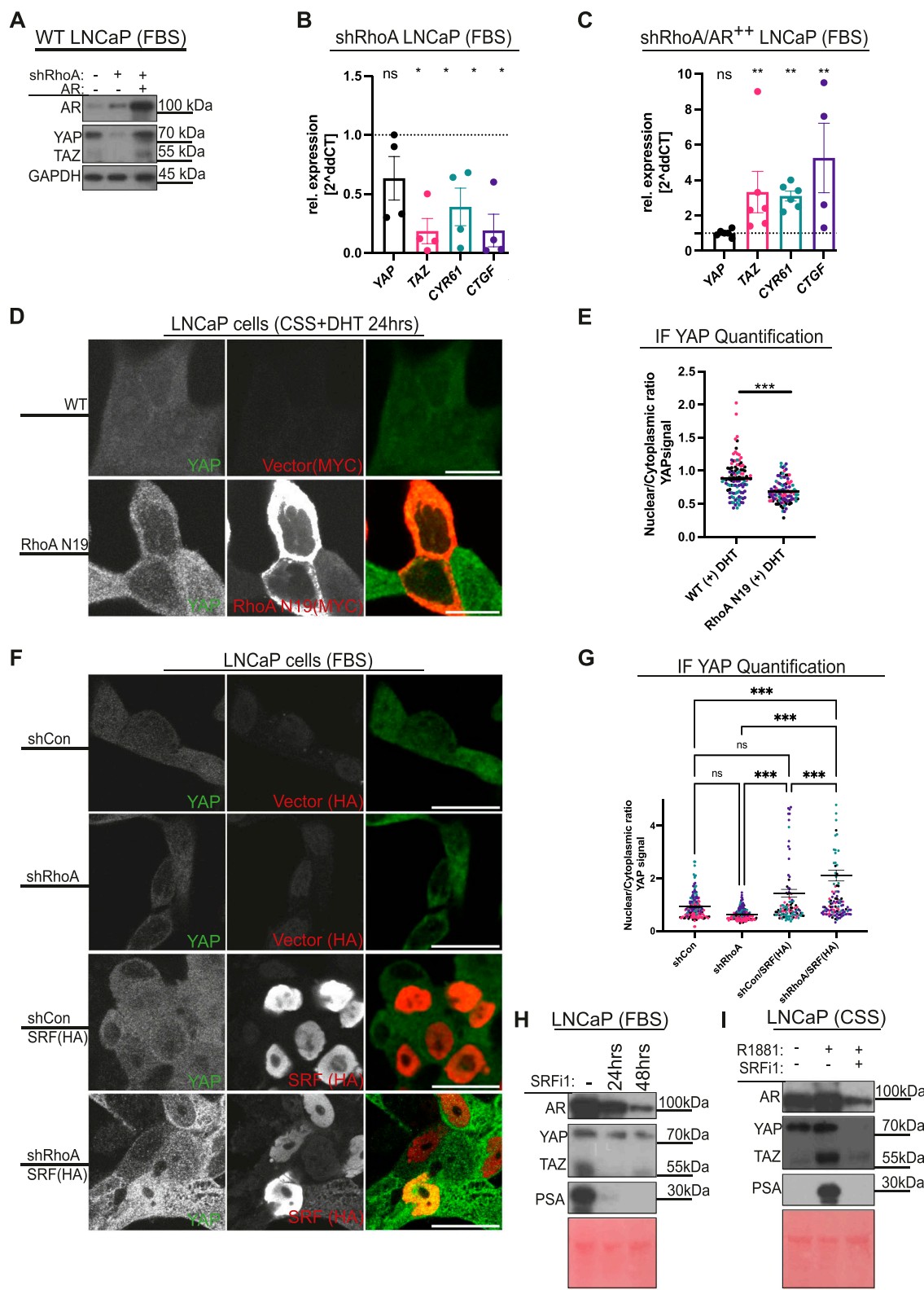

**Figure 3. AR-YAP/TAZ activation is regulated by RhoA-SRF.**

**(A)** Western blot analysis of LNCaP lysates from three different populations of cells, expressing short hairpin (sh) shCon LNCaP, targeting *RhoA* mRNA (shRhoA #1), and shRhoA #1 LNCaP exogenously expressing AR as illustrated. All cells were cultured in FBS. **(B)** qRT-PCR analysis of *YAP*, *TAZ*, *CYR61*, and *CTGF* mRNA from shRhoA LNCaP cells compared with shCon LNCaP cells. **(C)** qRT-PCR analysis of *YAP*, *TAZ*, *CYR61*, and *CTGF* mRNA from shRhoA LNCaP cells exogenously expressing AR, labelled shRhoA/AR[++], compared with shCon LNCaP cells. **(D)** Confocal images of LNCaP cells expressing empty vector and LNCaP cells expressing dominant negative RhoA N19 stimulated with

2010). We recently proposed that the mechanism present in breast cancer, where hormone-dependent estrogen receptor (ER)-mediated Hippo and YAP/TAZ regulation occurs, might be conserved in PCa via AR (Salem & Hansen, 2019). Therefore, we next asked whether AR regulation of YAP protein and *TAZ(WWTR1)* expression in PCa are SRF dependent.

Firstly, we examined whether SRF is regulated by RhoA and AR signalling. Our data indicate that *SRF* mRNA is down-regulated in response to RhoA knockdown (Fig S1A). Interestingly, *SRF* mRNA is rescued upon exogenous AR expression in RhoA-deficient LNCaP cells (Fig S1B). Consistent with the qRT-PCR analysis, we observed an increase in SRF protein upon DHT- or R1881-mediated AR stimulation, whereas knockdown of AR causes SRF protein loss (Fig S1C–F). To further dissect the RhoA-SRF regulation of YAP, we performed confocal-based IF imaging. Exogenous expression of HA-tagged SRF in LNCaP cells causes YAP nuclear localization, whereas RhoA knockdown favours YAP cytoplasmic localisation (Fig 3F and G). This cytoplasmic YAP localisation upon shRhoA is reversed by exogenously expressed SRF. Overall, this indicates that RhoA regulates YAP at least partly via SRF (Fig 3F and G).

In a complementary line of experiments, LNCaP cells cultured in media containing FBS were treated with the first generation SRF inhibitor, CCG1432 (SRFi 1) (Evelyn et al, 2007) for 24 and 48 h. SRF inhibition down-regulates YAP and TAZ protein levels (Fig 3H). Noteworthily, we found that inhibiting SRF reduces AR total protein and AR activity, as indicated by low PSA levels (Fig 3H). Furthermore, we next asked whether YAP/TAZ activation by AR is reversed in response to SRF inhibition. LNCaPs cultured in CSS conditions were either stimulated for 48 h with R1881, or simultaneously with R1881 and SRFi 1. SRF inhibition causes loss of AR-induced YAP, TAZ, and PSA proteins (Fig 3I).

To further confirm the results obtained using first generation SRF inhibitors, we took advantage of two different second generation SRF inhibitors with reported higher specificity, CCG222740 (SRFi 2) and CCG203971 (SRFi 3) (Yu-Wai-Man et al, 2017). We utilised a similar experimental set up as described above, and LNCaP cells were cultured in FBS conditions and treated with either SRF inhibitors independently. This SRF inhibition by these two compounds phenocopied SRFi 1, as they decreased YAP/TAZ total protein and lowered AR activity, as highlighted by diminished PSA levels (Fig S2A and B). Similarly, SRF inhibitors 2 and 3 cause loss of AR-mediated induction of YAP/TAZ and PSA (Fig S2C). To test whether SRF regulates *YAP/TAZ(WWTR1)* and their downstream activity, we performed qRT-PCR analysis on LNCaP cells cultured in CSS and treated for 48 h with either R1881 or R1881 and SRFi 2 or R1881 and SRFi 3. The addition of either of the SRF inhibitors in combination with R1881 caused a robust reduction in *TAZ* mRNA induction (Fig S2D). Furthermore, when we normalised LNCaP cells treated with R1881 to

LNCaP cells treated with R1881 and SRFi 2 or 3, respectively, we identified that *TAZ*, *CYR61*, and *CTGF*, *AR* and AR targets were suppressed (Figs S2E and F). However, no effect is observed on *YAP* mRNA levels (Figs S2E and F).

To ensure that the transcriptional inhibition of *TAZ(WWTR1)* is mediated via the specific activity of the SRF inhibitors used, we in complimentary experiments targeted *SRF* with shRNA. Consistently, *SRF* knockdown reduces *TAZ(WWTR1)* mRNA levels, whereas *YAP* mRNA remains unchanged (Fig S2G and H), thereby mirroring the data obtained using SRFis.

### YAP/TAZ are not essential for AR activation

The role of YAP as a mediator of CRPC has previously been examined. In these studies, knocking down *YAP* in the CRPC cell line C4-2 results in the loss of their hormone-insensitive phenotype (Zhang et al, 2015b). In addition, castrated mice treated with the YAP-TEAD inhibitor verteporfin (Liu-Chittenden et al, 2012) results in lower tumour growth rate (Jiang et al, 2017). Of note, the specificity in using verteporfin as a YAP inhibitor has come into question (Zhang et al, 2015a; Cunningham & Hansen, 2022). Consequently, how YAP/TAZ regulates AR activation in early/locally advanced hormone-sensitive PCa is not well understood.

To answer this question, and because of the general lack of specific YAP/TAZ inhibitors, we undertook a genetic approach and used CRISPR/Cas9 gene KO technology to generate YAP KO LNCaP cells (Fig 4A). These cells are consequently a valuable experimental tool, as compared with knockdown studies, no residual YAP is present. Next, we sought to confirm AR sensitivity in YAP KO LNCaP clones (Fig S3). The AR target genes *KLK2*, *PSA*, and *TMPRSS2* are robustly down-regulated in response to 24 h enzalutamide treatment in YAP KO cells (Fig S3A and B). In addition, PSA protein is reduced in these YAP-deficient cells in response to CSS treatment, which is rescued upon R1881 stimulation for 48 h (Fig S3C). We further confirmed that YAP loss does not affect LNCaP cell identity, as there is no robust change in the luminal epithelial cell markers, *CK18* and *CK8* upon YAP KO (Fig S3D). In addition, both YAP KO clones do not express the basal cell makers *CK14* and *TP63* (Fig S3D). Remarkably, TAZ protein and gene expression were abolished upon YAP loss in FBS conditions (Figs 4B and S3C). Importantly, the reintroduction of YAP in LNCaP YAP KO cells restores *TAZ(WWTR1)* levels (Fig 4C). We next asked whether *TAZ* gene activation by androgens is YAP-dependent. Two different LNCaP YAP KO clones cultured in CSS conditions were DHT-stimulated for 24 h, lysed, and analysed. These data reveal that AR-driven *TAZ* (*WWTR1*) gene transcription is conserved between WT LNCaP and YAP KO clones, however, increased TAZ protein expression is not detected by Western blotting (Fig 4D–F). This is consistent with the observation

---

androgen (10 nM, dihydrotestosterone, DHT) for 24 h in CSS conditions. YAP (green) and Myc-tagged RhoA (red). Brightness and contrast of the merged images were enhanced to allow the visualisation of the different channels. Scalebar = 15 μm. **(E)** Dot plot of quantified YAP nuclear/cytoplasmic levels, respectively, from images, as shown in (D). Colour-coding of dots represent cells analysed on the same day. **(F)** Confocal images of shCon, shRhoA, shCon/SRF(HA), and shRhoA/SRF(HA) LNCaP cells cultured in FBS for 24 h. YAP (green) and HA-tagged SRF (red). Mean ± SEM, Mann–Whitney *U* test. Brightness and contrast of the merged images were enhanced to allow the visualisation of the different channels. Scalebar = 30 μm. **(G)** Dot plot of quantified YAP nuclear/cytoplasmic levels, respectively, from images, as shown in (F). Mean ± SEM, two-way ANOVA. **(H)** Western blot analysis of lysates from LNCaP cells cultured in FBS conditions and treated with SRF inhibitor 1 (SRFi1, 10 μM, CCG1423) for 24 h, 48 h or vehicle (DMSO). **(I)** Western blot analysis of lysates from LNCaP cells cultured in CSS conditions. Cells were either left untreated or treated with androgen (1 μm, R1881) or combined androgen (1 μM, R1881) and SRFi1 (10 μM, CCG1423).

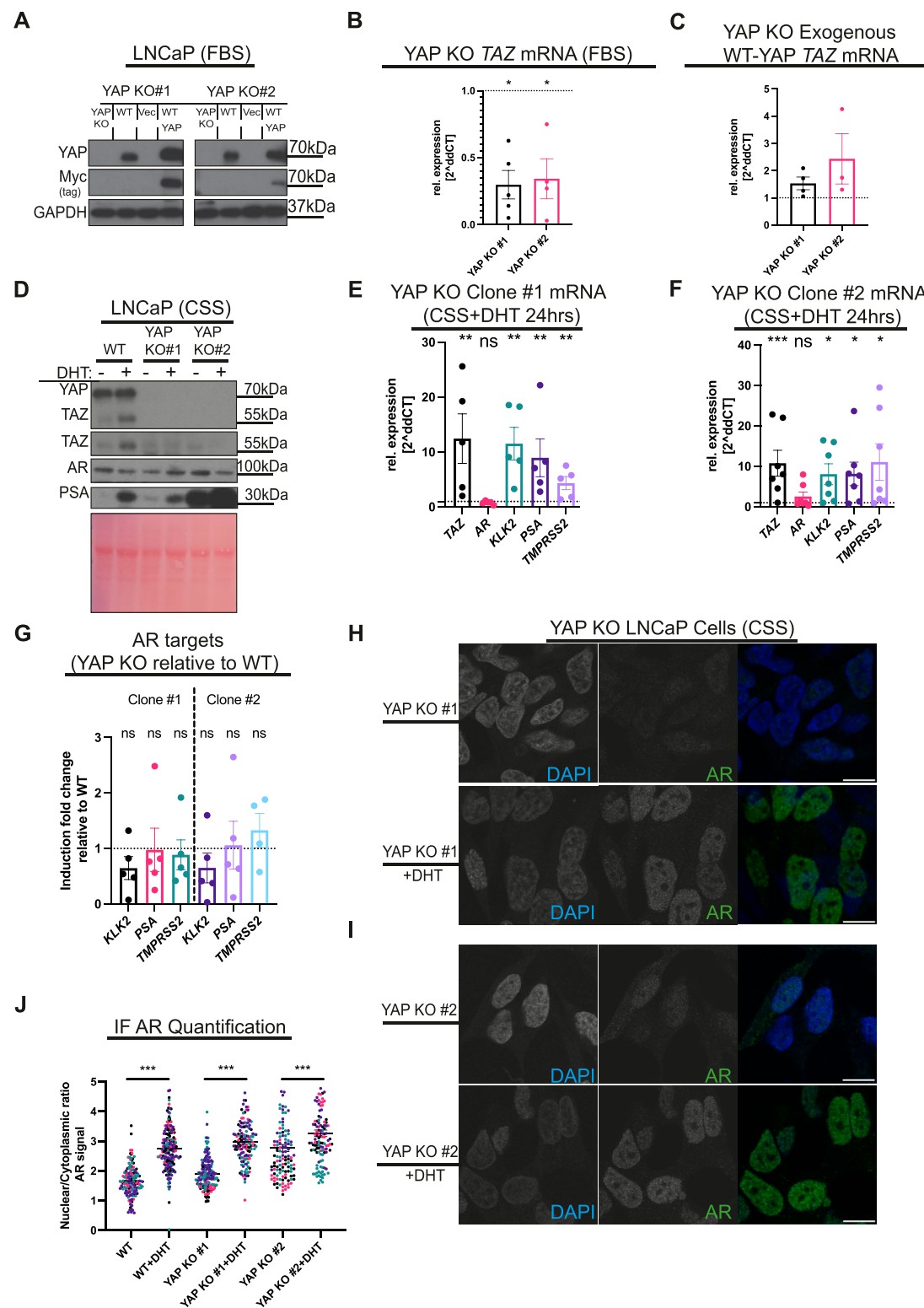

**Figure 4. YAP/TAZ is not essential for AR activation.**
**(A)** Western blot analysis of cellular lysates from LNCaP YAP KO clones #1 and #2, WT LNCaP, LNCaP YAP KO clones expressing vector control and LNCaP YAP KO clones expressing exogenous Myc-tagged WT-YAP. GAPDH serves as a loading control. **(B)** qRT-PCR analysis of *TAZ*(*WWTR1*) levels. mRNA from LNCaP YAP KO clones #1 and #2 compared with WT LNCaP cells cultured in FBS conditions. Each dot represents a biological replicate. Mean ± SEM, Mann–Whitney *U* test. **(C)** qRT-PCR analysis of *TAZ*(*WWTR1*). mRNA from LNCaP YAP KO clones #1 and #2 expressing exogenous WT-YAP compared with LNCaP YAP KO clones #1 and #2 cultured in FBS conditions. Each dot represents data from a biological replicate. Mean ± SEM. **(D)** Western blot analysis of lysates from WT LNCaP cells, LNCaP YAP KO clones #1 and #2. Cells were cultured in

that TAZ protein increase by R1881 stimulation is lost upon CHX-mediated translation inhibition (Figs 2D and S3C), a treatment that thereby phenocopies the YAP KO LNCaP clones. Taken together, the data highlight that *TAZ*(*WWTR1*) expression in CSS+DHT conditions is YAP-independent; however, TAZ protein translation only occurs when YAP protein is present (Figs 4D and S3C). We hypothesis that this YAP dependency is likely mediated via YAP transcriptionally regulated gene products. Because TAZ protein levels are dramatically decreased in response to YAP loss in LNCaP cells (Figs 4D and S4C), and the YAP KO LNCaP cell line is a robust model to study the role of both YAP and TAZ in regulating AR activation.

Next, we measured the expression of AR targets, *KLK2*, *PSA*, and *TMPRSS2*, in response to DHT exposure in the two YAP KO LNCaP clones. AR targets are robustly induced independently from YAP/TAZ (Fig 4E and F). Consistently, PSA protein is increased after DHT stimulation in both YAP KO clones (Fig 4D). Furthermore, we normalised the induction of AR targets in LNCaP YAP KO clones stimulated with DHT relative to WT LNCaP cells stimulated with DHT, which show no significant change upon YAP/TAZ genetic ablation (Fig 4G). AR translocates to the nucleus for its transcriptional activity (Simental et al, 1991; Zhou et al, 1994; Cutress et al, 2008; Lv et al, 2021), and YAP is reported to complex with AR in the nucleus, a complex deemed essential for AR downstream gene expression (Kuser-Abali et al, 2015). We consequently sought to firmly establish whether AR translocation to the nucleus upon androgen stimuli is YAP dependent using LNCaP cells engineered to not express YAP. Consistently, we found that YAP/TAZ loss in both YAP KO clones does not affect AR nuclear translocation in response to DHT treatment (Fig 4H–J). Thus, we establish that AR nuclear translocation and activation can occur independently from YAP/TAZ.

### Targeting AR–SRF–YAP/TAZ provides therapeutic potential

Our results provide insights into a novel signalling module in PCa, where AR activates RhoA-SRF in a feedforward manner, which induces YAP/TAZ activation. Each component of this pathway is a critical regulator of cancer hallmarks (Haga & Ridley, 2016; Antonarakis, 2018; Cunningham & Hansen, 2022; Azam et al, 2022). Hence, we hypothesize that targeting the AR–RhoA–SRF–YAP/TAZ signalling axis has therapeutic potential (Cunningham & Hansen, 2022). To assess the physiological and clinical relevance of our proposed regulatory mechanism between SRF and YAP/TAZ activity, we accessed a publicly available patient dataset, the prostate cancer transcriptomic atlas (http://www.thepcta.org) (You et al, 2016). We performed correlation analysis between *SRF* expression and the expression of *TAZ*(*WWTR1*), and the TAZ target genes *CYR61* and *CTGF*, respectively. Our analyses indicate that there is a robust positive correlation between *SRF* expression and the expression of *TAZ*(*WWTR1*), *CYR61* or *CTGF* ($\rho$ = 0.33–0.62) (Fig 5A–C). As expected, we did not record a correlation

between *SRF* and *glyceraldehyde 3-phosphate dehydrogenase* (*GAPDH*), a commonly used "house-keeping" gene (Fig 5D).

We were encouraged with these results and therefore proceeded to test the therapeutic feasibility of this proposed mechanism. Most nonmalignant cells receive positional signals from cell–cell interactions and the extracellular matrix to survive, as they otherwise undergo anoikis (Guadamillas et al, 2011). The ability to circumvent anoikis is a critical hallmark of cancer cells, as anoikis resistance allows cancer cells to expand, invade adjacent tissues, and ultimately to disseminate giving rise to metastasis (Guadamillas et al, 2011). Anchorage-independent growth is consequently a hallmark of most types of cancer development, (Pickup et al, 2014; Moroishi et al, 2015a; Crosas-Molist et al, 2022) and this ability of cancer cells is frequently driven by YAP/TAZ activity (Moroishi et al, 2015a). We used the soft agar colony formation assay to measure the ability of PCa cells' anchorage-independent growth in a 3D environment (Fig 5E). YAP/TAZ loss results in a substantial reduction in PCa colonies formed (Fig 5E and F). In addition, enzalutamide treatment results in less 3D colonies formed in both WT and YAP KO LNCaP cells (Fig 5E and F). Because AR is responsive to inhibition in YAP KO cells (Fig S4A and B), we asked whether there is an additive, and therefore potential synergistic effect, by combining chemical AR inhibition and YAP genetic targeting. The number of colonies upon treatment with enzalutamide is significantly down-regulated in YAP KO cells relative to WT LNCaP cells (Fig 5E and F). Notably, we did not observe an effect on 2D cell survival or 2D proliferation in response to YAP loss (Fig S4A and B). In addition, we did not detect a synergistic effect when we combined AR and YAP inhibition in 2D cultures (Fig S4C and D), excluding general changes in cell proliferation or cell death as the main mechanisms overcoming anoikis (Guadamillas et al, 2011). We speculate that YAP/TAZ modulates extracellular matrix deposition and expression of cell surface molecules in transformed PCa cells, which enhances their attachment and hence survival in unfavourable conditions. Finally, we identify that SRF inhibition suppresses anchorage-independent growth across both PCa genotypes, in agreement with our observation that SRF inhibition, inhibits both YAP and AR (Fig 3). Targeting SRF directly might therefore be a complimentary and potentially more effective treatment strategy than solely targeting AR or YAP.

## Discussion

YAP and TAZ act as responders and direct mediators of mechanical, metabolic, and signalling cues to promote cell growth and differentiation (Hansen et al, 2015a; Moya & Halder, 2018). Previous work showed that AR acts as a chaperon or a cargo protein that promotes YAP recruitment into the nucleus and increases its stability. YAP and AR complex formation occurs in a DHT-dependent manner in

---

CSS conditions for 48 h. Cells were treated with vehicle (DMSO) or androgen (10 nM, dihydrotestosterone, DHT) for 24 h. Ponceau S total protein stain serves as loading control. Note the loss of TAZ protein in LNCaP YAP KO clones. **(E, F)** qRT-PCR analysis of TAZ, AR, and established AR targets in LNCaP YAP KO clones #1 and #2. Each dot represents data from a biological replicate. Mean ± SEM Mann–Whitney *U* test. **(G)** Induction folds shown in (E, F) were normalised to induction folds shown in Fig 1F. Mean ± SEM, Mann–Whitney *U* test. **(H, I)** Confocal images of LNCaP YAP KO clones #1 and #2 treated with vehicle (DMSO) and androgen (10 nM, dihydrotestosterone, DHT) for 24 h in CSS conditions. DAPI (blue) and AR (green). Scalebar = 30 μm. Brightness and contrast of the merged images were enhanced to allow the visualisation of the different signals. **(J)** Dot plot of quantified AR cellular localisation, respectively, from images, as shown in (H). Each dot represents one cell. Cells from the same experiment are colour-coded. Note that LNCaP WT data are from the same experiment shown in Fig 1D, and shown here for comparison. Mean ± SEM, Mann–Whitney *U* test.

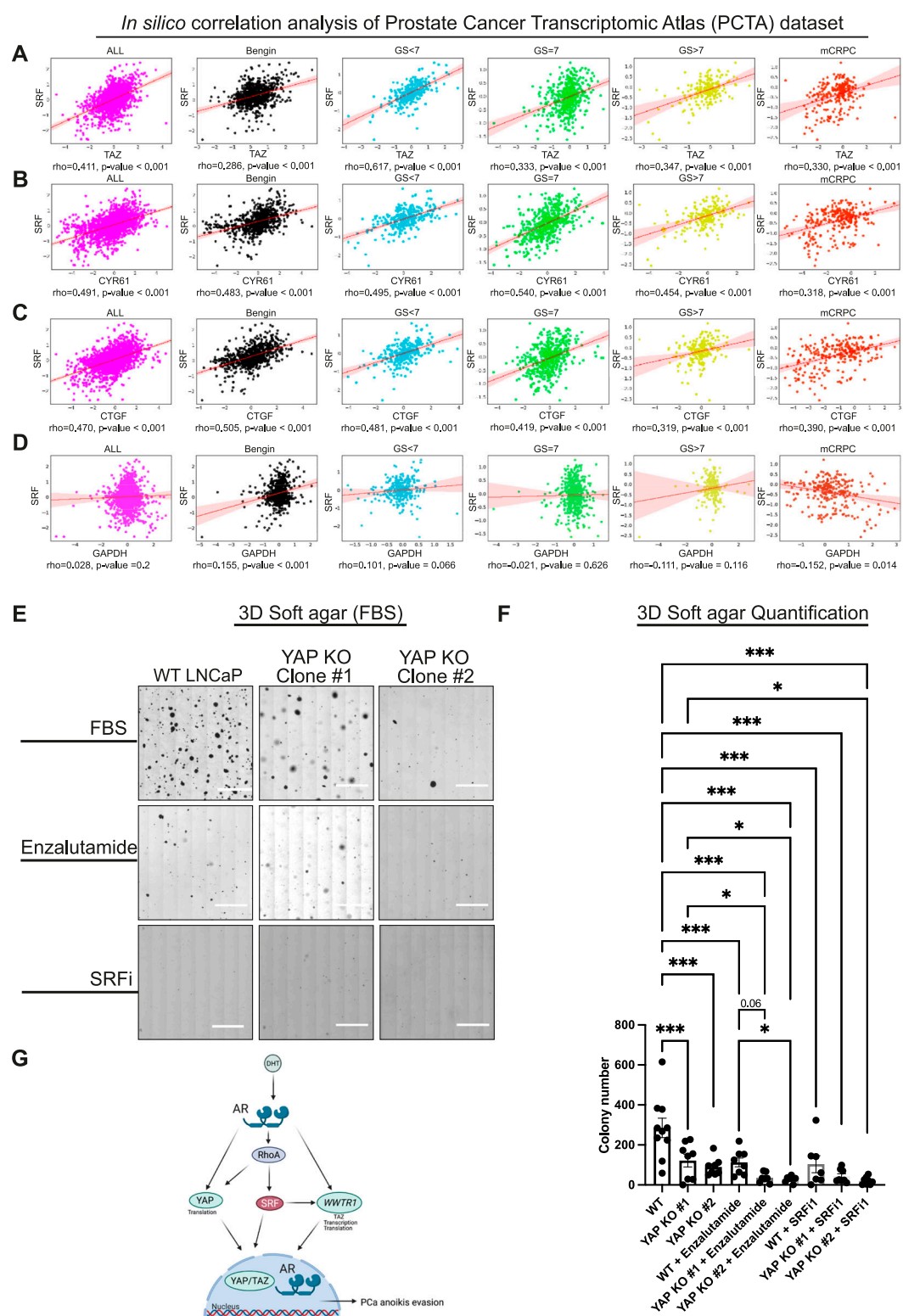

Figure 5. Targeting AR/YAP/SRF provides therapeutic potential.
(A, B, C, D) In silico correlation analysis of clinical data comparing *SRF* expression (A) with *TAZ*(*WWTR1*), (B) with *CYR61*, (C) with *CTGF* expression and (D) the house-keeping gene, *GAPDH* expression, respectively. Each dot represents the expression levels in an individual patient sample from the prostate cancer transcriptome atlas PCTA cohort (n = 2,115), which includes benign (benign prostatic hyperplasia, n = 794); primary prostate cancer (PCa) with Gleason Sum (GS) < 7 (n = 328), GS = 7 (n = 530), or GS > 7 (n = 203); and mCRPC samples (n = 260). (E) Representative images of 3D soft agar assay examining anchorage-independent growth. WT LNCaP and LNCaP YAP KO clones #1 and #2 were seeded in FBS containing soft agar and treated with vehicle (DMSO), AR antagonist (10 μM, enzalutamide) or SRFi1 (10 μM, CCG1423). Scalebar = 2.5 mm. Image brightness was enhanced to allow the visualisation of the colonies. (F) Soft agar quantification from images as in (E), mean ± SEM, one-way ANOVA.

hormone naïve cells; however, this interaction takes place independently from androgens in hormone-resistant cells (Kuser-Abali et al, 2015). Here, we delineate a new mechanism by which AR activates YAP and TAZ. We provide evidence that androgens promote YAP synthesis via AR, supporting YAP protein stability in hormone-naïve LNCaPs (Fig 1A and G). We speculate that this might be mediated via the eukaryotic initiation factor 4E (eIF4F) translation initiation complex (Liu et al, 2019). In comparison, *TAZ*(*WWTR1*) is transcriptionally induced upon androgen stimulation via AR directly binding to the region of the *WWTR1* promoter (Fig 2H). Although YAP and TAZ in general are thought to be regulated in a similar manner, our findings highlight the differential regulatory mechanisms controlling YAP and TAZ activity. YAP/TAZ activation via AR drives downstream gene expression, which promotes PCa anchorage-independent growth.

We here show that YAP and TAZ activation in response to AR occurs in a RhoA-SRF–regulated manner. SRF is a MADS box containing transcription factor (Onuh & Qiu, 2021). SRF induces early response genes after serum exposure and later targets that modulate the organisation of the actin cytoskeleton (Schratt et al, 2001). The canonical signalling pathway upstream of SRF is the RhoGTPase–actin–megakaryotic acute leukemia (MAL) axis (Miralles et al, 2003). RhoA activation facilitates actin polymerisation leading to the release of MAL from monomeric actin (Miralles et al, 2003). Active MAL translocates to the nucleus to induce SRF-mediated transcription (Miralles et al, 2003). RhoA also activates SRF via myocardian protein (MRTFA/B) (Gau & Roy, 2018). Our analysis reveals that androgens activate SRF in a feedforward manner. Consistently, unbiased transcriptomic analysis indicates that ~6% of the androgen-responsive genes in PCa are expressed independently from AR-ARE elements, but via the SRF binding motif CArG box, whereas 12% of SRF targets in PCa are androgen dependent (Heemers et al, 2011).

We show that RhoA–SRF acts as a signalling nexus between AR and YAP/TAZ; however, mechanistic insights into if a mutual dependence between MRTF-SRF and YAP/TAZ-TEAD in PCa cells take place is currently not fully understood. MRTF-SRF and YAP-TEAD can induce overlapping gene signatures to trigger changes in the actin cytoskeleton in cancer-associated fibroblasts (Foster et al, 2017). However, an alternative mechanism of action, where SRF and YAP/TAZ potentially act in parallel to induce separate gene signatures in response to androgens and thereby triggering the emergence of distinctive cellular types in the PCa niche might exist. This scenario was previously reported in breast cancer and epithelial cells (Kim et al, 2015b). In this context, SRF facilitates the recruitment and binding of YAP to the mammary stem cell signature–gene promoters independently from YAP-TEAD (Kim et al, 2015b).

The regulation of AR trafficking and subcellular localisation in the context of YAP/TAZ signalling in PCa is intriguing. AR activation can occur independently from YAP/TAZ (Fig 4). However, targeting YAP/TAZ sensitizes the cells to AR inhibition in 3D cultures (Fig 5E and F). This highlights a therapeutic window, as this combinatorial treatment targeting these transcriptional regulators would predominantly target transformed cells resistant to anoikis. We hypothesis that YAP/TAZ in PCa does not directly alter the genome wide AR-DNA interactions, but rather changes the cellular genetic state independently from AR.

Increasing evidence supports a non-AR-targeting strategy as an effective method to tackle AR resistance (Coffey, 2021). The Hippo pathway and YAP/TAZ are gaining increased attention as potential therapeutic candidates in PCa (Salem & Hansen, 2019; Coffey, 2021). A recent report showed that inhibitor of nuclear factor K B kinase subunit E (IKBKE) induces LATS2 turnover, resulting in YAP overexpression which mediates PCa tumorigenesis (Bainbridge et al, 2021). In this instance, targeting IKBKE results in overcoming AR resistance (Bainbridge et al, 2021).

Our data implicate AR, RhoA-SRF, YAP, and TAZ as major players in promoting PCa anchorage independent growth (Fig 5). Targeting SRF results in inhibition of AR and YAP/TAZ. Functionally, SRF or YAP inhibition causes a reduction in PCa cells' anchorage-independent growth ability, a critical feature of later stage cancer development. We complement and compare our extensive in vitro-based cellular analysis with patient data analysis extracted from prostate cancer transcriptomic atlas. This highlights that the findings observed in vitro are correlated with clinical datasets. Consistent with our findings, AR- and SRF-dependent gene expression is associated with PCa disease progression and lower survival rate (Schmidt et al, 2012). Future investigations are required to assess the therapeutic efficacy of inhibiting YAP/TAZ and SRF in hormone-naïve and castration-resistant PCa in preclinical and clinical settings.

# Materials and Methods

### Cell culture

Cell lines were cultivated at 37°C in a humidified 5% $CO_2$ atmosphere. LNCaP and C4-2 cells were cultured in RPMI-1640 media (Gibco) supplemented with 100 μg/ml penicillin/streptomycin (Lonza), 2 mM glutamine (Lonza), and 10% FBS. PC-3, DU145, HEK293T cells were cultured in high-glucose DMEM (Gibco) supplemented with 100 μg/ml penicillin/streptomycin, 2 mM glutamine Lonza, and 10% FBS.

For androgen stimulation assays, cells were cultivated in RPMI-1640 or DMEM supplemented with 10% CSS (Gibco), penicillin, streptomycin (Lonza), 2 mM glutamine (Lonza) for 48 h before the addition of 10 nM, DHT (Sigma-Aldrich) or 1 μM R1881 (Sigma-Aldrich).

---

**(G)** Graphical representation of proposed mechanism of YAP/TAZ activation by AR in an SRF-regulated manner. DHT binds to AR that induces translation of YAP, and the transcription of *WWTR1*, the gene encoding TAZ. Active AR and YAP/TAZ translocate to the nucleus to induce expression of established target genes and promote tumorigenesis. The molecular mechanism is not fully delineated. This proposed mechanism is regulated by SRF which regulates AR in a feedforward manner. Importantly, our evidence suggests that AR and SRF in prostate cancer is a regulator of YAP/TAZ levels and downstream activity.

## Chemicals

Compounds were purchased in powder form and resuspended in dimethyl sulfoxide (DMSO). DMSO was used as vehicle across all experiments. Compounds used are SRF inhibitors CCG-1423 (Merck), CCG-203971 (MedChemExpress), and CCG-222740 (MedChemExpress), enzalutamide (Enz) (Selleckchem), cyclohexamide (Cambridge Bio-Science), MG-132 (Sigma-Aldrich), DHT (Merck/Sigma-Aldrich), and R1881 (Merck/Sigma-Aldrich).

## Transformation and plasmid DNA preparation

Competent bacteria were transformed by heat shock. Bacteria were seeded onto carbenicillin-selective LB agar-containing Petri dishes. Single clones were picked the next day and propagated in overnight cultures for plasmid harvest.

## CRISPR-Cas9 YAP KO

Complementary guide oligo nucleotides were annealed and ligated into a linearised pSpCas9(BB)-2A-Puro (PX450 V2) plasmid, Addgene (#48139) at the BbsI restriction site, to generate the YAP KO construct. Guide RNA sequences targeting YAP were designed previously in Hansen et al (2015b) (forward primer; CACCGCAT-CAGATCGTGCACGTCCG and reverse primer; AAACCGGACGTGCAC-GATCTGATGC). Cells were transfected using Lipofectamine LTX transfection reagent (Invitrogen) following the manufacturer's protocol. Selection for the population of interest was performed for 48 h using puromycin (1 μg/ml) (AlfaAesar). Single-cell sorting was performed in 96-well plates containing RPMI supplemented with 20% FBS using the BD FACS Aria II (with the assistance of the QMRI flow cytometry team). Colonies were expanded and replicate plates were obtained, which were screened for positive KOs via Western blots. Whole uncut membranes of Western blots were used to ensure that no truncated versions of the proteins were observed. In addition, second validation Western blots using additional antibodies recognising distinct parts of the target proteins were performed (not shown). In addition, YAP rescue cells were generated, whereby YAP KO clones were transduced with pQCXIH-empty as a vector control or reexpressing myc-tagged WT-YAP pQCXIH-Myc YAP plasmid, Addgene (#33091).

## Retrovirus-mediated shRNA knockdown and transfection

HEK293T cells were co-transfected with pMD2G (200 ng/well) and pSPAX2 (400 ng/well) plasmids (from the Kun-Liang Guan lab, UCSD) with pKLO.1 vector (400 ng/μl) coding for shRNA targeting AR (TRCN0000350462 and TRCN0000350462) or RhoA (TRCNTRCN0000047710) or SRF (TRCN0000003981 and TRCN0000273541). GenJet was used as a transfection reagent. For the purpose of AR exogenous induction experiments, the AR plasmid, pLENTI6.3/AR-GC-E2325, Addgene (#85128) was used and pLEX 307 Addgene (#41392) was used as control. The media of transfected cells were harvested 48 h posttransfection and filtered through a low-binding syringe filter (Corning). Cells of interest were pretreated with fresh media containing 10 mg/ml polybrene. 50–100 μl of the virus solution was added to each well for 6–8h. 24 h after posttransduction, puromycin (1 μg/ml)

(AlfaAesar) was used for selection of the population of interest. For the purpose of IF experiments, RhoA KD cells and shCon cells treated were transfected with the dominant negative RhoA N19 plasmid Addgene (#15901) (Nobes & Hall, 1999) or SRF(HA) plasmid, pCGN-SRF, Addgene (#11977) (Johansen & Prywes, 1993) using Lipofectamine LTX transfection reagent (Invitrogen) following the manufacturer's protocol.

## RNA extraction and quantitative PCR

RNA isolation and extraction from cells was performed following the protocol provided in RNeasy plus mini kit (QIAGEN). RNA yields were then quantified using a Nano drop 1000 spectrophotometer (Thermo Fisher Scientific). 100 ng/μl RNA was used for cDNA synthesis following the protocol provided in SuperScript IV VILO Master Mix (Invitrogen). Quantitative reverse transcriptase–polymerase chain reaction (qRT-PCR) was performed in technical duplicates for the gene of interest using Brilliant III Ultra-Fast SYBR Green QPCR Master Mix (Agilent Technologies). The qRT-PCR reaction was performed using QuantStudio 5 Real-Time PCR System (Thermo Fisher Scientific). The mean CT value of the technical replicates was obtained and the expression of each gene of interest was normalised to the genetic expression of hypoxanthine–guanine phosphoribosyl-transferase (*HPRT1*). The control treatments were then normalised to 1 to determine the differential expression of each treatment. Each biological replicate (n = 3–7) was plotted in Prism (GraphPad) as a bar graph. The error bars represent the standard error mean. Sequences of the primers used are in Table 1.

## Western blot analysis

Cell lysates were prepared in reducing buffer and Western blots were performed as described in Hansen et al (2015b). The following antibodies: YAP (ab52771), YAP/TAZ (63.7/sc-101199), TAZ (V386/4883), PSA/KLK3 (D11E1), AR (N-20/sc-816), and AR (441/sc-7305) were used at a concentration of 1:1,000, MYC (9B11), GAPDH (sc-25778), HSP90 (BD610418) were used at a concentration of 1:5,000. Ponceau S (Sigma-Aldrich) was used as a loading control. Furthermore, anti-rabbit immunoglobulins/HRP P044801 and anti-mouse immunoglobulins/HRP P044701 secondary antibodies were used at a concentration of 1:10,000. Western blot membranes were developed using Immuno Western ECL mix (Millipore) and x-ray films (SLS).

## IF microscopy

IF slides and cover slips were prepared as described in Rausch et al (2019). Primary antibodies recognising YAP (EP1674Y), YAP/TAZ (63.7/sc-101199), AR (N-20/sc-816), and HA Tag (6E2/2367S), and Alexa Flour 488- and 594- conjugated secondary antibodies (Invitrogen) were used. Cover slips were mounted on slides using ProLong Diamond Antifade Mount with DAPI stain (Thermo Fisher Scientific). Image acquisition was performed using Leica TCS SP8 confocal laser scanning microscope utilising the HC PL AP Oil 63x1.4 CS2 objective. The nuclear-to-cytoplasmic signal intensity ratio was quantified in 5–15 cells of each population from between 8–15 images. The nucleus was identified in the DAPI channel and used to

**Table 1.** qRT-PCR primer sequence.

| Target | Forward primer (5′-3′) | Reverse primer (5′-3′) |
|---|---|---|
| *HPRT* | AGAATGTCTTGATTGTGGAAGA | ACCTTGACCATCTTTGGATTA |
| *YAP* | CCAAGGCTTGACCCTCGTTTT | TCGCATCTGTTGCTGCTGGTT |
| *TAZ (WWTR1)* | AATGGAGGGCCATATCATTCGAG | GTCCTGCGTTTTCTCCTGTAT |
| *CYR61* | AGCCTCGCATCCTATACAACC | TTCTTTCACAAGGCGGCACTC |
| *CTGF* | CCAATGACAACGCCTCCTG | TGGTGCAGCCAGAAAGCTC |
| *KLK2* | CCTGGCTTCCGCAACTTACAC | GGACTTGTGCATGCGGTACTCA |
| *PSA (KLK3)* | TATTGTAGTAAACTTGGAACCTTG | TTACACCATTTAAGAAACACTCTG |
| *TMPRSS2* | CTGCCAAGGTGCTTCTCATT | CTGTCACCCTGGCAAGAATC |
| *AR* | CCTGGCTTCCGCAACTTACAC | GGACTTGTGCATGCGGTACTCA |
| *RhoA* | AGCCTGTGGAAAGACATGCTT | TCAAACACTGTGGGCACATAG |
| *SRF* | CGAGATGGAGATCGGTATGGT | GGGTCTTCTTACCCGGCTTG |
| *TP63* | GGACCAGCAGATTCAGAACGG | AGGACACGTCGAAACTGTGC |
| *CK14* | GAGATGTGACCTCCTCCAGC | TCAGTTCTTGGTGCGAAGGA |
| *CK18* | ATCTTGGTGATGCCTTGGAC | CCTGCTTCTGCTGGCTTAAT |
| *CK8* | TAGCACTGGGAACAGGAGA | TTTGACATTGGCAGAGCTA |

draw the region of interest. The channel of interest was picked, and the signal intensity of the nucleus was measured and tabulated. The region of interest was further moved to the cytoplasm, measured, and tabulated to determine the signal intensity outside the nucleus. The value obtained from the nucleus was excluded from the latter images and the nuclear-to-cytoplasmic ratio was quantified in Microsoft Excel and plotted as a scatterplot in GraphPad Prism.

### Soft agar assay

2x RPMI was supplemented with 20% FBS, 2% penicillin/streptomycin (Lonza) and 2% L-glutamate (Gibco) was prepared. For the bottom layer, 1% agar (BD Biosciences) was prepared in sterile dH$_2$O and mixed 1:1 with 2X complete RPMI, poured in six well plates and left to solidify. For the top layer, 5,000 cells were added to 0.7% agar in sterile dH$_2$O mixed with 2x RPMI supplemented with 1.7 g/l NaHCO$_3$ (CarlRoth). 1 ml of 1x complete RMPI containing either the vehicle (DMSO) or an inhibitor of interest was added on top of the solid agar. Cells were cultured for 10–14 d at 37°C and media were changed once a week. Cells were fixed in 0.005% crystal violet (Thermo Fisher Scientific) in methanol (Fisher chemicals) overnight. Methanol was used to destain the agar. Images of colony abundance were obtained by EVOS FL Auto two-cell imaging system using the 20X objective/magnification (Thermo Fisher Scientific). The number of colonies (of at least 50 pixels in diameter) per well was quantified using Fiji (ImageJ). The macro used for quantification is in Table 2.

**Table 2.** ImageJ Macro used for counting soft agar colonies.

```
run("8-bit");
run("Subtract Background…," "rolling = 50 light");
setOption("BlackBackground," false);
run("Make Binary");
run("Convert to Mask");
run("Watershed");
setAutoThreshold("Default");
//run("Threshold…");
call("ij.plugin.frame.ThresholdAdjuster.setMode," "B&W");
setThreshold(129, 255);
//setThreshold(129, 255);
run("Convert to Mask");
run("Analyze Particles…," "size = 50-Infinity pixel circularity = 0.250–1.00 show = Outlines display exclude clear summarize");
```

## Cell proliferation assay (MTT tetrazolium reduction assay)

For each experiment, cells were seeded in 96-well plates at a starting density of 2–5 x $10^4$ cells per well in (n = 3) technical replicates. Cells were seeded in (n = 3) biological replicates, in full FBS media, full CSS media, DMSO (vehicle treated) or a titrated concentration of enzalutamide (Selleckchem). Cells were harvested on days 0, 1, 3, and 5 d. 50 $\mu$l serum free media and 50 $\mu$l of MTT solution were added to each well. Plates were incubated for 4 h at 37°C. 100 $\mu$l of MTT solvent (Abcam) were added into each well. The plate was wrapped in foil and placed on an orbital shaker for 15 min. Absorbance was read at OD = 570. Differential change in absorbance was plotted across the experimental period as a measure for cell proliferation or used to calculate cell survival percentage.

## Dataset analysis

PCa cell line ChiP-Seq data were extracted and analysed from the Cistrome database to visualise potential AR binding to the *WWTR1* and *YAP1* promoter regions. The Cistrome data are from LHSAR treated with R1881 for 24 h (GSM1716768), C4-2 (GSE71704), LNCaP treated with R1881 for 24 h (GSM1527823), and VCaP cells treated with R1881 for 2 h (GSM1354831).

## Quantification and statistical analysis

Data were analysed using Fiji (ImageJ), Excel (Microsoft), and Prism (GraphPad). All statistical analyses were performed using Prism software. Significance is represented using (p) value, "*" $P \leq 0.05$, "**" $P \leq 0.01$, and "***" $P \leq 0.001$ or "ns" for nonsignificant difference. Statistical significance of the data were tested and analysed using Mann–Whitney $U$ test, unless stated otherwise. Data are represented as mean ± SEM as highlighted in the figure legends. Super plots were used to visualise the reproducibility between the different IF experiments (Lord et al, 2020); each cell was used as a data point for statistical analysis. Colourblind safe GraphPad Prism format was used across the manuscript.

# Supplementary Information

# Acknowledgements

Work ongoing in the Gram Hansen laboratory was supported by a University of Edinburgh Chancellor's Fellowship and by Worldwide Cancer Research (19-0238) and LifeArc-CSO. Additional funding has been obtained from the Wellcome Trust–University of Edinburgh Institutional Strategic Support Funds (ISSF), ISSF2, and ISSF3. This research was therefore funded, in part, by the Wellcome Trust [Grant number 204804/Z/16/Z]. For the purpose of open access, the author has applied a CC BY public copyright licence to any author accepted manuscript version arising from this submission. S Jia is funded by a scholarship from the Chinese Scholarship Council, and the Edinburgh Global from University of Edinburgh. We furthermore acknowledge team members for helping with cell culturing and for insightful comments on this study. We acknowledge the technical support and guidance provided by the Centre for Reproductive Health SuRF Histology, Imaging and qRT-PCR Facility staff, in addition to the QMRI Flow Cytometry and Cell Sorting Facility staff. Team members within the Gram Hansen laboratory are thanked for their constructive feedback during this study.

## Author Contributions

O Salem: data curation, formal analysis, validation, investigation, visualization, methodology, and writing—original draft, review, and editing.
S Jia: data curation, formal analysis, validation, investigation, visualization, methodology, and writing—review and editing.
B-Z Qian: resources, supervision, and writing—review and editing.
CG Hansen: conceptualization, resources, data curation, formal analysis, supervision, investigation, methodology, and writing—original draft, review, and editing.

## Conflict of Interest Statement

The authors declare that they have no conflict of interest.

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
