## [Reviewer comments · Life Science Alliance]

Life Science Alliance

AR activates YAP/TAZ differentially In Prostate Cancer Cells

Omar Salem, Siyang JIA, Bin-Zhi Qian, and Carsten Hansen

DOI: <https://doi.org/10.26508/lsa.202201620>

Corresponding author(s): Carsten Hansen, University of Edinburgh

Review Timeline:

Submission Date:	2022-07-20
Editorial Decision:	2022-08-26
Revision Received:	2023-05-04
Editorial Decision:	2023-05-24
Revision Received:	2023-06-12
Accepted:	2023-06-13

Scientific Editor: Novella Guidi

Transaction Report:

August 26, 2022

Re: Life Science Alliance manuscript #LSA-2022-01620-T

Dr. Carsten G Hansen
University of Edinburgh
MRC Center for Inflammation Research
Institute for Regeneration and Repair, Queen's Medical Research Institute, Edinburgh bioQuarter, 47 Little France Crescent
Edinburg
United Kingdom

Dear Dr. Hansen,

Thank you for submitting your manuscript entitled "AR activates YAP/TAZ differentially In Prostate Cancer Cells" to Life Science Alliance. The manuscript was assessed by expert reviewers, whose comments are appended to this letter. We invite you to submit a revised manuscript addressing the Reviewer comments.

Thank you for this interesting contribution to Life Science Alliance. We are looking forward to receiving your revised manuscript.

Sincerely,

B. MANUSCRIPT ORGANIZATION AND FORMATTING:

Reviewer #1 (Comments to the Authors (Required)):

Summary:

In this manuscript, Salem et al. address the mutual regulatory interactions between androgen receptor (AR) signalling and the paralogous transcriptional coregulators YAP and TAZ that drive prostate tumorigenesis. They show that the AR regulates YAP at the level of protein translation, while it regulates TAZ at the level of transcription. Both effects are regulated by signalling mechanism involving RhoA and SRF. In prostate cancer patients, SRF expression was found to correlate positively with expression of the canonical YAP/TAZ target genes CYR61 and CTGF. The authors round up their manuscript by showing that targeting YAP/TAZ or SRF sensitizes prostate cancer cells to AR inhibition. Overall, this paper will be of interest to the (prostate) cancer and Hippo/YAP/TAZ signalling readerships, and I recommend publication pending careful attention to the comments below.

Scope/novelty:

While the interaction of YAP/TAZ with the AR in the nucleus of prostate cancer cells (including the LNCaP cells used in this study by Salem et al.) has been thoroughly characterised (Kuser-Abali, G., Alptekin, A., Lewis, M. et al. YAP1 and AR interactions contribute to the switch from androgen-dependent to castration-resistant growth in prostate cancer. *Nat Commun* 6, 8126 (2015). <https://doi.org/10.1038/ncomms9126>), the signalling mechanisms resulting from this interaction have not. Of note, the study of Kuser-Abali did not investigate the roles of TAZ, nor did it study the regulatory feedback interactions between YAP and TAZ. The effects of RNAi-mediated YAP knock-down or pharmacological inhibition (using Verteporfin, whose specificity as a YAP/TAZ inhibitor is heavily debated) on the proliferation of prostate cancer cells in 3D growth conditions have already been shown by the study from Kuser-Abali et al. That said, the synergistic effect of YAP genetic knock-out and AR inhibition shown in this manuscript is novel. It is also worth highlighting that Salem used CRISPR/CAS9 technology to generate YAP knockout cells. Consequently, both studies combined allow for a more solid conclusion about the roles of YAP and TAZ in prostate tumorigenesis. I would therefore like to believe that the manuscript merits publication in Life Science Alliance.

Experimental design and data quality:

The manuscript is, for the most part, technically sound. The experiments are well designed and controlled. For most parts of the results section, the data were analysed appropriately and include sufficient repeats to allow statistical testing and interpretation. While the overall technical quality of this manuscript is okay, it is, unfortunately, not high enough at this stage for me to recommend publication. In particular, the statistical analysis should be improved in a several cases (which I have indicated in my general comments below) to make some of the claims more convincing.

Major comments:

- The manuscript is difficult to read in parts, and it will greatly benefit from careful editing, as there are a several typos and spelling/grammar mistakes that necessitate a thorough revision
- Given that YAP/TAZ function as both, co-activators and co-repressors of gene transcription, they should be referred to as transcriptional co-regulators.
- Statistics: There is mentioning of the statistical tests used for data analyses, but four-star statistical significance (Figure 1) is not explained in the methods section.

For larger data sets, where tests performed to see if the data showed normal distribution? And where statistical tests then performed based on a normal or not-normal data distribution?

For the analyses of qPCR data (e.g. Fig. 1E etc), one-way Anova with post-hoc testing for multiple comparisons might be the better statistical test, as it helps to answer the question if the various mRNAs analysed followed a similar trend.

- Quantification of immunofluorescence data; Figure 1B-D, Figure 3D-G, Figure 4H-J: It appears to me that the dot plots are showing pooled data from multiple experiments (no information is provided about how many biological replicates were analysed). This would have led to high statistical significance while, for example, the immunostainings in panel B show only modest increase in nuclear YAP/TAZ in response to (DHT) treatment. If the data plots do indeed show pooled data, then the data should ideally be presented as super plots that communicate both the cell-level variability and the experimental reproducibility (<https://pubmed.ncbi.nlm.nih.gov/32346721/>), and statistical tests should be done over the means of the different biological replicates.

- Western blots: Was there a particular reason for using crops of Ponceau red-stained membranes as loading control? If so, I'd prefer if the authors displayed the whole stained membrane. However, a western blot of a housekeeping gene (whose

expression is stable across the different conditions) is still the better loading control.

The contrast and resolution of the western blot panels could be improved.

- Figure 1A: YAP levels appear to correlate with AR levels, while AR activity levels (as assessed by PSA expression) correlate with TAZ expression.
- Figure 2C: it appears that CHX treatment has some effect on TAZ expression. Could the authors please comment on this?
- Figure 2 H and I: could the authors please highlight the promoter regions better?
- Figure 3: a western blot is missing that shows efficient knock-down of RhoA upon shRNA treatment.
- Figure 4B: it would make more sense to combine the two graphs into one. This would allow the reader to more readily appreciate that re-expression of WT-YAP in YAP knock-out cells rescues TAZ expression.
- Page 11 and Figure 4G: contrary to what is stated in the text, the figure doesn't show changes in PSA expression.
- Figure 5F: the claim that Enzalutamide treatment results in less 3D colonies in YAP KO compared to WT cells is not supported by the way the data are analysed, as no comparison (using a statistical test) is made between the respective groups.
- Figure S1D: the increase in SRF expression in WT cells upon R1881 treatment is barely visible.
- Figure S4A-B: the data presented do not show that 'AR (expression?) is responsive to (Enzalutamide) inhibition in YAP KO cells', but that the expression of AR target genes is.

Minor comments:

- Figures 2A, Figure S1F and S4C: the labelling is confusing and inconsistent. E.g. it not easy to grasp that in lane 6 in Figure 2A cells were grown in CCS conditions for 48h, then treated for another 4h hours with R1881, and then treated with MG132 for 6 hours. related to changes in subcellular distribution of proteins.
- In Figure 2, panel C should come after panel D, to be consistent with the flow of the results text.
- Page 5, second line: I think the sentence should read: 'Notably, the INCREASED expression...'
- Page 7: Figures 2H and I are referred to as Figures 3H and I.
- Page 11, first paragraph. I guess the third sentence should read: Our results show that AR-driven TAZ gene TRANSCRIPTION is conserved between WT LNCaP and YAP KO clones, however, INCREASED TAZ protein EXPRESSION was not detected by western blotting?
- Page 13, first paragraph: why is Figure S4 referred to before Figure S3?

Reviewer #2 (Comments to the Authors (Required)):

This interesting manuscript shows that Androgens, which play a key role in prostate cancer, can activate YAP/TAZ, by increasing their levels either transcriptionally or post-translationally. Furthermore, the activation of YAP/TAZ is dependent on the RhoA-regulated signal MRTF-SRF. Finally, YAP KO prostate cancer cells grow poorly in soft agar. Some correlations across prostate cancer transcriptomic sets are also shown. Overall, the manuscript supports the notion that YAP/TAZ play an oncogenic role in prostate cancer and may contribute to the potent action of androgens in this tumour type. I support publication with minor revisions.

Comments.

1. In the abstract, there are some typographical errors of singular/plural that should be corrected.
2. The immunostaining for YAP is often rather dim in the figures, so should be enhanced.

Reviewer #3 (Comments to the Authors (Required)):

The current manuscript evaluates the role of AR in regulating YP/TAZ levels and activity. This manuscript provides evidence for a role of YAP/TAZ in AR dependent prostate cancer with potential therapeutic implications.

- 1) The authors did an excellent job in demonstrating the role of AR in regulating YAP and TAZ however the data justifying the role of RhoA as an intermediary in this process is not as well developed and the data is not overly convincing that RhoA is the dominant mediator. It seems that AR can regulate YAP/TAZ in the absence of RhoA and unlikely plays a dominant role. Furthermore, the authors show no direct effect of AR on RhoA in this context.
- 2) In vivo therapeutic studies should be performed along with demonstration of target inhibition employing either KO/KD strategies or therapeutic inhibitors (SRFi2) +/- AR inhibition.
- 3) the in vitro assays for the experiments performed should also be conducted in other AR positive cell lines such as CWR22PC, VCaP, or prostate cancer organoids.
- 4) mechanistic understanding of how the prostate cancer biology is influenced by YAP/TAZ in this AR dependent context would

be useful. Are there any differences in cell identify/plasticity/proliferation/EMT/ migration, etc...

Dear Editor and Reviewers

We thank you for your time and constructive feedback.

We have now completed our studies with additional experimental and edited the manuscript accordingly, with additional analyses to strengthen the findings within. We believe these additions have resulted in a more complete picture of our studies and the interactions between the molecular players, which we feel together have improved the manuscript overall. We have also included additional references in order to provide further context. The revision experiments took longer than expected, as the lead author moved onto new (greener?) pastures. We thank you for your patience, while we have conducted our revisions.

We hope you are now satisfied with the work, and like us, feel it is ready for publication.

We here below go through the reviewer comments. Our response in red.

Reviewer #1 (Comments to the Authors (Required)):

Summary:

In this manuscript, Salem et al. address the mutual regulatory interactions between androgen receptor (AR) signalling and the paralogous transcriptional coregulators YAP and TAZ that drive prostate tumorigenesis. They show that the AR regulates YAP at the level of protein translation, while it regulates TAZ at the level of transcription. Both effects are regulated by signalling mechanism involving RhoA and SRF. In prostate cancer patients, SRF expression was found to correlate positively with expression of the canonical YAP/TAZ target genes CYR61 and CTGF. The authors round up their manuscript by showing that targeting YAP/TAZ or SRF sensitizes prostate cancer cells to AR inhibition. Overall, this paper will be of interest to the (prostate) cancer and Hippo/YAP/TAZ signalling readerships, and I recommend publication pending careful attention to the comments below.

Scope/novelty:

While the interaction of YAP/TAZ with the AR in the nucleus of prostate cancer cells (including the LNCaP cells used in this study by Salem et al.) has been thoroughly characterised (Kuser-Abali, G., Alptekin, A., Lewis, M. et al. YAP1 and AR interactions contribute to the switch from androgen-dependent to castration-resistant growth in prostate cancer. *Nat Commun* 6, 8126 (2015). <https://doi.org/10.1038/ncomms9126>), the signalling mechanisms resulting from this interaction have not. Of note, the study of Kuser-Abali did not investigate the roles of TAZ, nor did it study the regulatory feedback interactions between YAP and TAZ. The effects of RNAi-mediated YAP knock-down or pharmacological inhibition (using Verteporfin, whose specificity as a YAP/TAZ inhibitor is heavily debated) on the proliferation of prostate cancer cells in 3D growth conditions have already been shown by the study from Kuser-Abali et al. That said, the synergistic effect of YAP genetic knock-out and AR inhibition shown in this manuscript is novel. It is also worth highlighting that Salem used CRISPR/CAS9 technology to generate YAP knockout cells. Consequently, both studies combined allow for a more solid conclusion about the roles of YAP and TAZ in prostate tumorigenesis. I would therefore like to believe that the manuscript merits publication in Life Science Alliance.

Experimental design and data quality:

The manuscript is, for the most part, technically sound. The experiments are well designed and controlled. For most parts of the results section, the data were analysed appropriately and include sufficient repeats to allow statistical testing and interpretation. While the overall technical quality of this manuscript is okay, it is, unfortunately, not high enough at this stage for me to recommend publication. In particular, the statistical analysis should be improved in a several cases (which I have indicated in my general comments below) to make some of the claims more convincing.

We thank the reviewer for his/her time, detailed and overall, very supportive feedback.

Major comments:

- The manuscript is difficult to read in parts, and it will greatly benefit from careful editing, as there are a several typos and spelling/grammar mistakes that necessitate a thorough revision.

We have now carefully edited the manuscript, including by having a native English speaker help edit our writing. This has improved the readability. We thank the reviewer for his/her comment.

- Given that YAP/TAZ function as both, co-activators and co-repressors of gene transcription, they should be referred to as transcriptional co-regulators.

We now highlight that YAP/TAZ can both repress and activate distinct gene-sets and include references for this. This section now reads “Notably, YAP and TAZ when binding to TEAD and potentially other transcription factors both activates and repress specific gene-sets, and as such can both function as either gene-specific co-activators or co-repressors. When the upstream kinase module of the Hippo pathway is engaged, this leads to LATS1/2 mediated YAP and TAZ phosphorylation on multiple serine sites.”. We have also included a range of relevant papers highlighting this as references. We have also changed the wording to co-transcriptional regulators in the abstract.

- Statistics: There is mentioning of the statistical tests used for data analyses, but four-star statistical significance (Figure 1) is not explained in the methods section.

For larger data sets, where tests performed to see if the data showed normal distribution? And where statistical tests then performed based on a normal or not-normal data distribution? For the analyses of qPCR data (e.g. Fig. 1E etc), one-way Anova with post-hoc testing for multiple comparisons might be the better statistical test, as it helps to answer the question if the various mRNAs analysed followed a similar trend.

We thank the reviewer for his/her comment. We have now conducted additional repeats, and we are not using four stars in the current figures. This level of significance claim is not relevant for our studies.

Anderson-Darlin Normality tests was performed for all IF experiments and the results revealed that the data are nonparametric, except in Figure 1C, where we therefore used

Students t-test. Statistical significance of other figures in the manuscript were tested by either Mann-Whitney t-test or Two-Way Anova.

In all qPCR experiments in the manuscript, we are comparing the mRNA fold change between two different treatments. We therefore use the non-parametric Mann-Whitney t-test to test for significance based on this experimental design, as we are not comparing multiple treatments.

- Quantification of immunofluorescence data; Figure 1B-D, Figure 3D-G, Figure 4H-J: It appears to me that the dot plots are showing pooled data from multiple experiments (no information is provided about how many biological replicates were analysed). This would have led to high statistical significance while, for example, the immunostainings in panel B show only modest increase in nuclear YAP/TAZ in response to (DHT) treatment. If the data plots do indeed show pooled data, then the data should ideally be presented as super plots that communicate both the cell-level variability and the experimental reproducibility (<https://pubmed.ncbi.nlm.nih.gov/32346721/>), and statistical tests should be done over the means of the different biological replicates.

We thank the reviewer for his/her comments. We have now carried out more experiments, and are now portraying the dot plots with different colours depending on the experimental day using colour safe GraphPad Prism format as “super plots”. These data show that our experiments are comparable across experimental days, and we are consequently using the individual cells as data points.

- Western blots: Was there a particular reason for using crops of Ponceau red-stained membranes as loading control? If so, I'd prefer if the authors displayed the whole stained membrane. However, a western blot of a housekeeping gene (whose expression is stable across the different conditions) is still the better loading control. Instead of relying on a specific house-keeping protein, we use various ways to confirm equal loading for our Western Blots. In some of early experiments we used HSP90 together with Ponceau as a dual control for equal loading. However, as HSP90 as discussed in our introduction is involved in the regulation of AR, and although we didn't observe a marked difference in HSP90 expression across genotypes, we prefer not to use these blots, and sought to use alternative ways to establish equal loading. We now include as requested larger sections of the membranes for the Ponceau stains (which provide signal from all proteins) or GAPDH levels (for instance figure 1J-L, as our loading controls).

The contrast and resolution of the western blot panels could be improved.

Thanks, we agree that some of these were not of good enough quality. We have now improved this, where we felt this was needed.

- Figure 1A: YAP levels appear to correlate with AR levels, while AR activity levels (as assessed by PSA expression) correlate with TAZ expression. In the manuscript we argue that AR regulates YAP protein stability, while binding to the *WWTR1* promoter to induce TAZ gene expression.

- Figure 2C: it appears that CHX treatment has some effect on TAZ expression. Could the authors please comment on this?

The reviewer is correct. However, we find that this is a modest effect on TAZ compared to YAP.

- Figure 2 H and I: could the authors please highlight the promoter regions better?

Thanks for the comment, this is now done. It is difficult to indicate where the promoter exactly is, however, we detect clear peaks in *WWTR1(TAZ)* predicted to most frequently being the regions of a gene specific promoter. We therefore highlight the approximate region of where we think the promoter lies, in order to not make substantial claims, of which we don't have the full experimental data for.

- Figure 3: a western blot is missing that shows efficient knock-down of RhoA upon shRNA treatment.

This is now included in Figure S1C.

- Figure 4B: it would make more sense to combine the two graphs into one. This would allow the reader to more readily appreciate that re-expression of WT-YAP in YAP knock-out cells rescues TAZ expression.

We prefer to keep these separate, as we feel this works better with the flow of the manuscript, as well as the figure representation.

Additionally, data presented in figure 4C are from cells that express plasmids from an empty vector or WT-YAP. Cells in figure 4B don't.

- Page 11 and Figure 4G: contrary to what is stated in the text, the figure doesn't show changes in PSA expression.

The reviewer is correct, this is a typological error, and the text should have referred to figure 4D, which is a WB that shows an increase in PSA expression in both YAP KOs. This is now changed to "4D" on page 12. We thank the reviewer for spotting this.

- Figure 5F: the claim that Enzalutamide treatment results in less 3D colonies in YAP KO compared to WT cells is not supported by the way the data are analysed, as no comparison (using a statistical test) is made between the respective groups.

- We have added additional repeats, so we now have in total > six biological replicates.

We have performed a two-way annova to compare WT LNCaP cells treated with Enzalutamide cells to LNCaP YAP KO clones treated with Enzalutamide cells.

- Additionally, we have performed "calculation for coefficient of drug interaction" (CDI)=0.832, which reveals synergy between Enzalutamide treatment and YAP knockout.

$CDI = AB / (A \times B)$.

Where AB is the ratio of the combination groups to control group
A or B is the ratio of the single agent group to control group.

Thus, CDI value <1, = 1 or >1 indicates that the drugs are synergistic, additive or antagonistic, respectively.

Treatment	Average colony number
WT LNCaP untreated	272
YAP KO LNCaP untreated (mean from 2 clones)	106
WT LNCaP + Enzalutamide	101
YAP KO LNCaP + Enzalutamide (mean from 2 different clones)	33

Formula	Ratio	
A	YAP Knockout : WT	0.389322917
B	WT + Enzalutamide : WT	0.371323529
AB	YAP knockout + Enzalutamide : WT untreated	0.120371586
A X B		0.14456476

$CDI = AB / (A \times B)$	0.832648195
--------------------

We have, however not included these CDI calculations in the current manuscript, as we are not certain that it adds further value to the current manuscript.

- Figure S1D: the increase in SRF expression in WT cells upon R1881 treatment is barely visible.
Thanks for the comment. This finding is not essential, and the phenotype although reproducible is not essential or dramatic. This figure is therefore not included in the current manuscript.
- Figure S4A-B: the data presented do not show that 'AR (expression?) is responsive to (Enzalutamide) inhibition in YAP KO cells', but that the expression of AR target genes is. This is a good point, and we have reworded this statement accordingly.

Minor comments:

- Figures 2A, Figure S1F and S4C: the labelling is confusing and inconsistent. E.g. it not easy to grasp that in lane 6 in Figure 2A cells where grown in CCS conditions for 48h, then treated for another 4h hours with R1881, and then treated with MG132 for 6 hours. related to changes in subcellular distribution of proteins.

We have updated the representation of the figure, which hopefully makes the figure easier to read.

- In Figure 2, panel C should come after panel D, to be consistent with the flow of the results text.

The order of the text has now changed

- Page 5, second line: I think the sentence should read: 'Notably, the INCREASED expression...'

The reviewer is correct, this is now updated

- Page 7: Figures 2H and I are referred to as Figures 3H and I.

The reviewer is correct, this is now updated.

- Page 11, first paragraph. I guess the third sentence should read: Our results show that AR-driven TAZ gene TRANSCRIPTION is conserved between WT LNCaP and YAP KO clones, however, INCREASED TAZ protein EXPRESSION was not detected by western blotting?

The reviewer is correct, this is now updated.

- Page 13, first paragraph: why is Figure S4 referred to before Figure S3?

The order of the figures as well as the text is now updated to make sure S3 is referred to first before S4.

We thank the reviewer for his/her time and helpful feedback

Reviewer #2 (Comments to the Authors (Required)):

This interesting manuscript shows that Androgens, which play a key role in prostate cancer, can activate YAP/TAZ, by increasing their levels either transcriptionally or post-translationally. Furthermore, the activation of YAP/TAZ is dependent on the RhoA-regulated signal MRTF-SRF. Finally, YAP KO prostate cancer cells grow poorly in soft agar. Some correlations across prostate cancer transcriptomic sets are also shown. Overall, the manuscript supports the notion that YAP/TAZ play an oncogenic role in prostate cancer and may contribute to the potent action of androgens in this tumour type. I support publication with minor revisions.

We thank the reviewer for his/her supportive feedback. Much appreciated!

Comments.

1. In the abstract, there are some typographical errors of singular/plural that should be corrected.

We have now corrected this, and thank the reviewer for his/her helpful feedback

2. The immunostaining for YAP is often rather dim in the figures, so should be enhanced.

We have at places implemented this, but overall want to make sure that we keep the image representations sound.

We thank the reviewer for his/her helpful feedback and supportive comments.

Reviewer #3 (Comments to the Authors (Required)):

The current manuscript evaluates the role of AR in regulating YP/TAZ levels and activity. This manuscript provides evidence for a role of YAP/TAZ in AR dependent prostate cancer with potential therapeutic implications.

1) The authors did an excellent job in demonstrating the role of AR in regulating YAP and TAZ however the data justifying the role of RhoA as an intermediary in this process is not as well developed and the data is not overly convincing that RhoA is the dominant mediator. It seems that AR can regulate YAP/TAZ in the absence of RhoA and unlikely plays a dominant role. Furthermore, the authors show no direct effect of AR on RhoA in this context.

Thanks for your enthusiasm regarding us demonstrating the role of AR in regulating YAP and TAZ. It is true that we don't have the full mechanistic insights into RhoA's role in this process, and that exogenous (forced) expression of AR circumvents shRhoA. However, knock down of RhoA cause diminished *TAZ*, *CTGF*, *CYR61* and *SRF* (figure 3B, C and S1A, B). In addition, expression of the dominant negative RhoA (N19) mutant blocks the androgen mediated YAP nuclear translocation (figure 3D, E).

We include these observations in the proposed working model in figure 5G.

2) In vivo therapeutic studies should be performed along with demonstration of target inhibition employing either KO/KD strategies or therapeutic inhibitors (SRFi2) +/- AR inhibition.

We compare our extensive in vitro based data with patient data from PCTA. We highlight in the discussion, that our data is not yet shown in vivo, but that our data positively correlates with clinical datasets. This section includes our final statement "*Future investigations are required to assess the therapeutic efficacy of inhibiting YAP/TAZ and SRF in hormone naïve and castration resistant PCa in preclinical and clinical settings*". Including in vivo data is outside the scope of the manuscript.

3) the in vitro assays for the experiments performed should also be conducted in other AR positive cell lines such as CWR22PC, VCaP, or prostate cancer organoids.

In silico analysis of AR binding to *TAZ/WWTR1* across a range of AR positive cell lines is shown in figure 2H.

4) mechanistic understanding of how the prostate cancer biology is influenced by YAP/TAZ in this AR dependent context would be useful. Are there any differences in cell identify/plasticity/proliferation/EMT/ migration, etc...

We thank the reviewer for this comment. These are interesting questions. We show that YAP (and TAZ) are needed for PCa cells for anchorage independent growth (figure 5E,F), but not for AR signalling (figure 4). Of note, this last finding is substantial, as it is at odds with previous findings, where researchers used knockdown studies or inhibitors to target YAP and TAZ. Knockout cells are a clean way to assess if a gene is essential for a cellular function. We have also analysed the expression of both Epithelial (CK8 and CK18) and Basal (CK14 and TP63) markers in YAP KO cells and compared these to WT LNCaPs. YAP KO do not increase expression of the epithelial markers (Figure S3D). Although LNCaP cells are considered epithelial, we only pick up low levels of *CK14* and *TP63* expression in WT cells, however these levels are trending lower upon YAP loss (Figure S3D).

We thank the reviewer for their time and feedback.

Overall

We thank the reviewers and the editor for their patience, comments and constructive feedback. After implementing these changes, through both experiments, and editing the manuscript as highlighted above, we now feel the paper is ready for publication.

May 24, 2023

RE: Life Science Alliance Manuscript #LSA-2022-01620-TR

Dr. Carsten G Hansen
University of Edinburgh
Center for Inflammation Research
Institute for Regeneration and Repair
Edinburgh BioQuarter, 4-5 Little France Drive
Edinburg EH16 4UU
United Kingdom

Dear Dr. Hansen,

Thank you for submitting your revised manuscript entitled "AR activates YAP/TAZ differentially In Prostate Cancer Cells". We would be happy to publish your paper in Life Science Alliance pending final revisions necessary to meet our formatting guidelines.

- please address the final Reviewer 1's comment
- please include your summary blurb in your main manuscript text after the title page
- please consult our manuscript preparation guidelines <https://www.life-science-alliance.org/manuscript-prep> and make sure your manuscript sections are in the correct order
- please use the [10 author names, et al.] format in your references (i.e. limit the author names to the first 10)
- please upload your Tables in editable .doc or excel format; -Tables should be numbered consecutively with Arabic numerals (1, 2, 3, 4); They can be included at the bottom of the main manuscript file or be sent as separate files.
- please upload your main and supplementary figures as single files;
- please add ORCID ID for corresponding (and secondary corresponding) author--you should have received instructions on how to do so
- please add your main, supplementary figure, and table legends to the main manuscript text after the references section

A. FINAL FILES:

B. MANUSCRIPT ORGANIZATION AND FORMATTING:

Sincerely,

Reviewer #1 (Comments to the Authors (Required)):

The authors have done a thorough job in addressing my comments, especially in improving data quality and interpretation, as well as the readability of the manuscript. I feel the manuscript now merits publication in Life Science Alliance.

Just one little comment related to Figures 2H and I and to the author's response to my comment: I would tone down the statement and maybe write something like 'These analyses indicate the presence of AR chromatin IP peaks in the DNA region presumably containing the TAZ promoter.'

Reviewer #3 (Comments to the Authors (Required)):

The authors have responded adequately to the reviewers original comments, and have appropriately strengthened their manuscript.

No further comments.

Dear Reviewers and Editor,

Reviewer #1 (Comments to the Authors (Required)):

The authors have done a thorough job in addressing my comments, especially in improving data quality and interpretation, as well as the readability of the manuscript. I feel the manuscript now merits publication in Life Science Alliance.

Just one little comment related to Figures 2H and I and to the author's response to my comment: I would tone down the statement and maybe write something like 'These analyses indicate the presence of AR chromatin IP peaks in the DNA region presumably containing the TAZ promoter.'

Thank you for your comment, we have now added "These analyses indicate the presence of IP peaks in the DNA region presumably containing the *TAZ (WWTR1)* promoter in a range of prostate cancer cell lines (LHSAR, C4-2, LNCaP, and VCaP) (Figure 2H)" when discussing this topic.

Reviewer #3 (Comments to the Authors (Required)):

The authors have responded adequately to the reviewers original comments, and have appropriately strengthened their manuscript.

Thank you.

Overall, we have also corrected a few wordings and typos.

Thank you very much for taking the time to provide feedback on our manuscript.

June 13, 2023

RE: Life Science Alliance Manuscript #LSA-2022-01620-TRR

Dr. Carsten G Hansen
University of Edinburgh
Center for Inflammation Research
Institute for Regeneration and Repair
Edinburgh BioQuarter, 4-5 Little France Drive
Edinburg EH16 4UU
United Kingdom

Dear Dr. Hansen,

Thank you for submitting your Research Article entitled "AR activates YAP/TAZ differentially In Prostate Cancer Cells". It is a pleasure to let you know that your manuscript is now accepted for publication in Life Science Alliance. Congratulations on this interesting work.

DISTRIBUTION OF MATERIALS:

Again, congratulations on a very nice paper. I hope you found the review process to be constructive and are pleased with how the manuscript was handled editorially. We look forward to future exciting submissions from your lab.

Sincerely,
